# Comparison of the most likely low-emission electricity production systems in Estonia

**Zachariah Steven Baird**[1]*, **Dmitri Neshumayev**[1], **Oliver Järvik**[1], **Kody M. Powell**[2]

**1** Department of Energy Technology, Tallinn University of Technology, Tallinn, Estonia, **2** Department of Chemical Engineering, University of Utah, Salt Lake City, Utah, United States of America

* zachariah.baird@taltech.ee

**Data Availability Statement:** All data is available in the Open Science Framework repository (https://osf.io/8b2dw/) and code used for the analyses is available on Github (https://github.com/zmeri/electricity-sim-estonia).

## Abstract

To meet targets for reducing greenhouse gas emissions, many countries, including Estonia, must transition to low-emission electricity sources. Based on current circumstances, the most likely options in Estonia are renewables with energy storage, oil shale power plants with carbon capture and storage (CCS), or the combination of renewables and either oil shale or nuclear power plants. Here we compare these different scenarios to help determine which would be the most promising based on current information. For the comparison we performed simulations to assess how various systems meet the electricity demand in Estonia and at what cost. Based on our simulation results and literature data, combining wind turbines with thermal power plants would provide grid stability at a more affordable cost. Using nuclear power to compliment wind turbines would lead to an overall levelized cost of electricity (LCOE) in the range of 68 to 150 EUR/MWh (median of 103 EUR/MWh). Using oil shale power plants with CCS would give a cost between 91 and 163 EUR/MWh (median of 118 EUR/MWh). By comparison, using only renewables and energy storage would have an LCOE of 106 to 241 EUR/MWh (median of 153 EUR/MWh).

## Introduction

Estonia has the goal of being a climate neutral economy by 2050 [1]. There is currently active discussion about what strategy Estonia should take to reduce greenhouse gas emissions to reach that goal at an acceptable price. The largest source of carbon emissions in Estonia is the oil shale industry, and in particular the oil-shale-fired power plants [2]. Oil shale is a type of fossil fuel that has been the main source of electricity in Estonia for decades and has enabled Estonia to have more energy independence. However, there is increasing pressure to limit its use. Even Eesti Energia, the company that operates the oil shale power plants, has pledged to stop producing electricity from oil shale by 2030 and to be carbon neutral by 2045 [3]. The four most likely strategies that could be used to achieve these goals and reduce the carbon emissions of Estonia's electricity sector are

- Capture the $CO_2$ from the oil shale plants

- Replace oil shale plants with renewables and a nuclear power plant

**Funding:** The Estonian Research Council (https://www.etag.ee/en/) funded this research under the National Programme for Addressing Socio-Economic Challenges through R&D (RITA), which is supported by the Estonian Government and European Regional Development Fund, under the project "Climate Change Mitigation with CCS and CCU Technologies" (ClimMit, grant No. RITA1/02-20). The funders had no role in study design, data collection and analysis, decision to publish, or preparation of the manuscript.

**Competing interests:** The authors have declared that no competing interests exist.

- Replace oil shale plants with renewables and energy storage

- Replace some oil shale plants with renewables and capture $CO_2$ from the remaining ones

Renewable energy sources now have costs low enough to compete with fossil fuels, but wind and solar power are intermittent and require additional infrastructure to achieve a stable grid. Onshore wind turbines are currently the cheapest low-emission energy source for Estonia. However, there is only a limited number of suitable locations on land in Estonia and often locals do not want them built near them. Offshore wind turbines, although more expensive, avoid some of these issues and can take advantage of the higher wind power density in the Baltic Sea [4]. Due to the relatively poor solar resources and higher installation costs in Estonia, solar panels are less effective and cost more than wind turbines [5]. In recent years the cost of renewable energy sources, such as wind turbines and solar panels, has significantly decreased. Now the cost of electricity from these sources is generally on par with that of fossil fuel power plants [5], which makes them an attractive alternative to oil shale. Indeed, a state-owned energy company, Enefit Green, is pursuing development of a 1100 MW offshore wind farm 12 km off the coast of Hiiumaa [6, 7]. This large installation would likely produce about 2.6 TWh of electricity per year, which is significant when considering that Estonia's annual demand is only 8.4 TWh [8, 9]. However, wind and solar are intermittent and cannot consistently meet demand on their own or maintain important grid parameters, such as the frequency.

The electricity supply could be stabilized if wind turbines and solar panels were coupled with a nuclear power plant, energy storage, or a smaller number of oil shale plants fitted with carbon capture. Carbon capture and storage is being investigated as a potential solution and there is also active discussion in Estonia about whether or not to construct a nuclear power plant [10–12]. Although Estonia can import electricity from neighboring countries to somewhat compensate for variation in the production from wind turbines, this is not a realistic substitute for having domestic dispatchable power because relying on imports would reduce Estonia's energy independence and might lead to an increase in prices. Additionally, many neighboring countries may also decommission their fossil fuel plants, meaning that there may not be a stable supply in neighboring countries that Estonia could rely on for imports to stabilize its own grid. Therefore, it is likely that Estonia would need to pair wind and solar power with a dispatchable form of electricity generation or storage.

Here we compare these various potential energy systems by performing simulations to see to what extent they are able to meet demand. Based on the simulation results we also estimate the levelized cost of electricity in each of these systems.

## Methodology

To study the performance of a variety of electricity systems in Estonia, we developed code to simulate the electricity demand and production from various sources. Functions for modeling electricity storage were also included. Each simulation covered a full calendar year with data calculated at 10 second intervals. Our code is written in Python and C++ and is available online (https://github.com/zmeri/electricity-sim-estonia). The code relies on the Scipy [13], Numpy [14], Pandas [15, 16], and Matplotlib [17, 18] packages.

To account for the uncertainty in the parameters underlying these simulations, we chose to use a Monte Carlo method. In this method the expected range for key parameters was described using distributions. In each simulation, parameter values were randomly selected from these distributions. The exact parameters used for these distributions can be seen from the code used for our simulation. Then the simulation was run 2000 or 3000 times (depending on the analysis) to get a distribution for each of the summary metrics we used to assess the

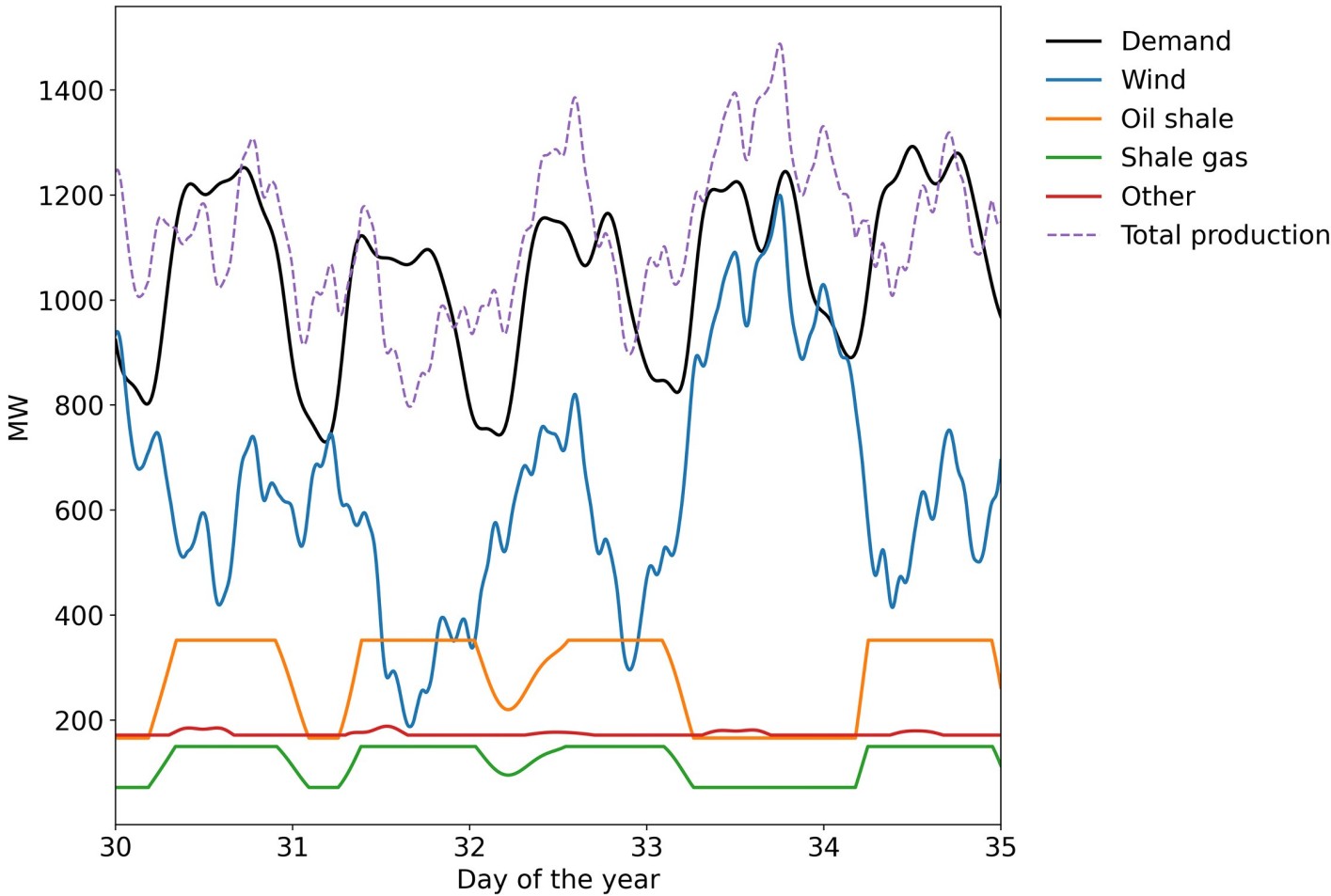

**Fig 1. Example of data from a simulation for electricity demand and production in Estonia.**

performance of the specified electricity system. This enables the uncertainty of the results to be taken into account and quantified, which can allow us to determine if a difference between any two scenarios is significant.

Representative data for a few days of a simulation is shown in Fig 1. More details about the different elements of the simulation are given in the following subsections.

## Demand

We used a discrete cosine transform (DCT) to model historical electricity demand data in Estonia, and then we selected similar coefficients and transformed them using an inverse DCT to generate realistic demand data for the simulation. Historical data from the five years of 2016–2020 was obtained from Elering's website [8]. By performing the DCT on each year separately, we obtained a set of 5 values for each variable in frequency space. From this set of values we were able to calculate the mean and standard deviation of each variable. To generate new demand data for the simulations, we simply selected a value for each variable in frequency space from the normal distribution described by its mean and standard deviation. We only used the first 5000 cosine terms because including additional terms made no significant difference in the result. Then, we transformed these generated values back using the inverse DCT to

obtain demand data for the simulation. The DCT functions in Scipy [13, 19] were used to perform these transformations. For more details, see our code that was used to implement the simulation.

Generated data was used instead of actual historical data because using actual data for a single year or even a few years does not capture the full range of variation that might occur in the future and could bias the results. The demand for a given day can vary quite a bit depending on the year. This is shown in Fig 2, which shows the historical data along with an example of generated demand data. For instance, on Day 234 the demand for 2020 was significantly lower than that of the previous 4 years. Such variations might be because the weather on a given day can be different from year to year or because some social factors could influence it (as an example, see how water usage in Canada changed significantly during a gold medal hockey game [20]). If only actual historical data is used, then that might introduce a bias into the simulation results because the full range of potential variation is not being represented.

We visually observed the generated demand data for several simulations and found it to replicate actual demand data quite well. It followed the patterns expected in the Estonian electricity grid, such as a higher electricity demand during the cold dark winter months and a lower demand during the summer [9, 21]. The model was even able to replicate patterns that are not immediately obvious, for instance the consistently low usage of electricity on the 23rd and 24th of June in Estonia.

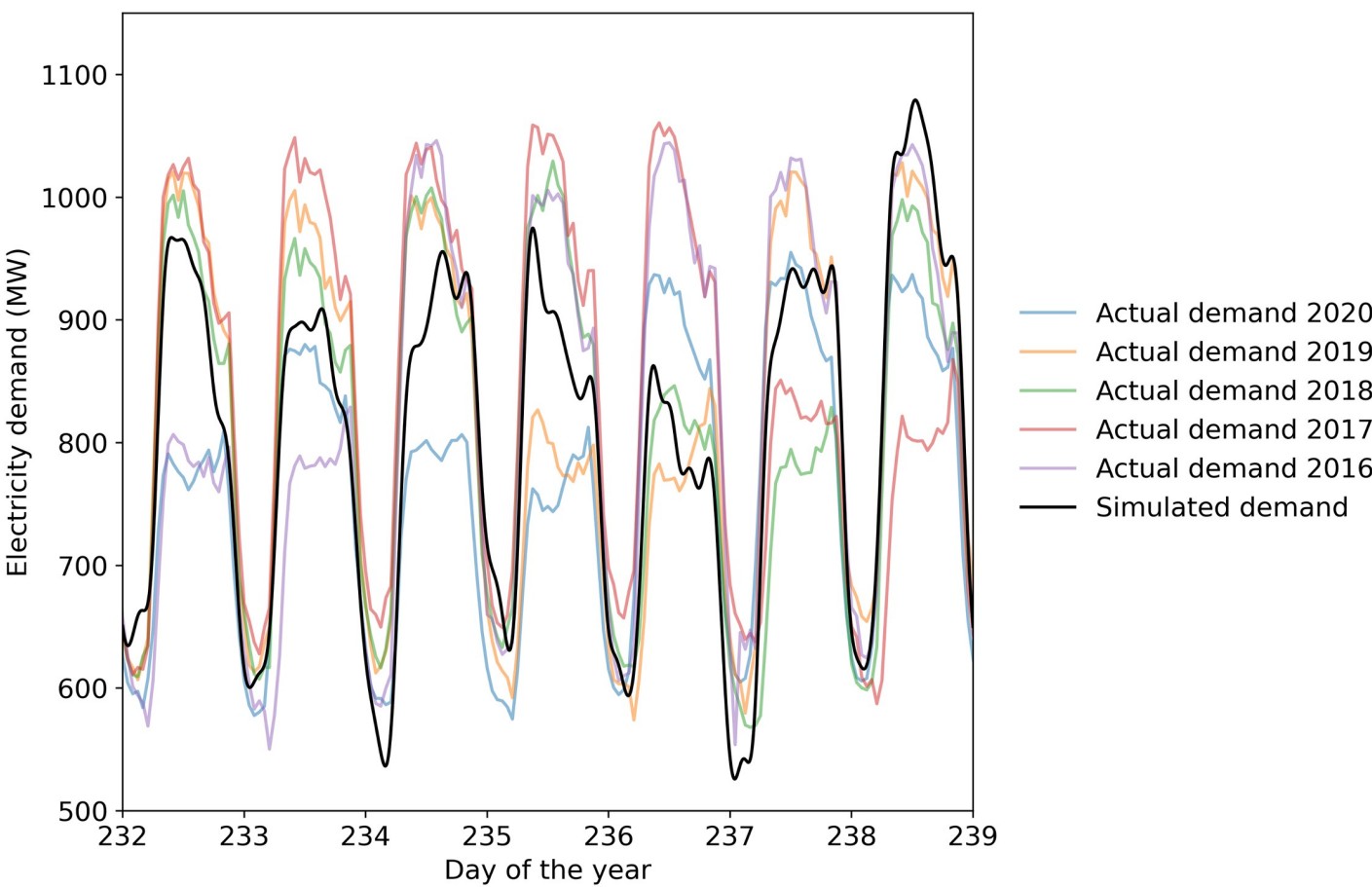

**Fig 2. Comparison of actual demand data and one example of generated demand data.**

## Production

We considered electricity production from combinations of the following sources:

- Wind turbines

- Solar panels

- Hydroelectric power plants

- Biomass and waste power plants

- Oil shale power plants with CCS

- Pyrolysis gas plants with CCS

- Nuclear power plants (generation III+ small modular reactor)

- Energy storage systems (pumped hydro storage)

Other fossil fuels were not considered because oil shale is locally available and importing other fossil fuels would reduce energy independence.

The capacity for hydropower, biomass, and waste was essentially kept the same for all the simulations (i.e. they were always selected from the same distributions). The annual mean capacity for these sources was set to be the same as available in 2019, as reported by Statistics Estonia [22, 23]. For biomass and waste power plants, this was equal to 160 MW and for hydropower it was 6 MW. Although biomass and waste are generally combusted in combined heat and power plants in Estonia, to our knowledge they generally still produce year-round. Data on wood fuel usage also indicates this by showing that wood fuel is still used during the summer months and the dip is not so large [24]. For this reason we kept production from these sources constant throughout the year.

Because wind and solar resources are intermittent and depend on the weather and solar irradiation, the capacity factors for these sources were modeled using an inverse DCT, implemented in Scipy [13, 19]. This allowed us to model the variability of these resources.

For the wind capacity factor we used the same method as used for modeling demand. First, actual wind production data for Estonia was obtained from Elering's website [8]. A DCT was performed on each year separately, and from these results the mean and standard deviation of each transformed variable could be calculated. These statistics were then used to calculate a normal distribution from which new values could be selected to generate wind production data for the simulation. For wind we only used the first 50,000 cosine terms because this was enough to accurately represent wind production for a year. An inverse DCT was applied to then obtain the wind capacity factor over time.

For the solar capacity factor, the first term in the series was set so that the average annual capacity factor matched the actual value for Estonia. The average capacity factor for solar panels was estimated to be 11% using the solar potential for Estonia given by the World Bank Group's Global Solar Atlas [25]. The coefficients for additional cosine terms were selected to give output during the day and no output at night. To take into account the effect of weather on solar output, we randomly selected a factor for each day to multiply by the capacity factor. This allowed us to simulate the large differences between neighboring days due to weather events, such as rain, clouds, or snow, which more closely resembles the trends for real solar panels [26].

Because oil shale and nuclear power plants are dispatchable, it was easier to model those. The output of such power plants is still limited by various issues, such as operational problems and the need for maintenance, and we took this into account by multiplying the nameplate

capacity by a capacity factor. The distributions for the capacity factors were chosen based on data in a National Energy Technology Laboratory report [27]. In recent years, the capacity factors for coal and oil shale plants have shown a decreasing trend, and we expect this is largely due to market and regulatory constraints that make their operation less economical [28–30]. We assumed that these constraints would be alleviated if CCS was added to the oil shale plants because CCS would reduce emissions and because some sort of governmental financial support would probably be needed for CCS to even be constructed. Therefore, we selected an 82% capacity factor for the oil shale plants based on the data from NETL [27]. Nuclear power plants generally have a capacity factor of about 93% [30].

Both oil shale and nuclear power plants can be operated flexibly, i.e., their output can be reduced during times when less electricity is needed. It may come as a surprise to some that nuclear power plants can also be operated flexibly, but EDF in France has operated its 58 reactors flexibly for more than 30 years. These plants have demonstrated the ability to ramp between 20% and 100% capacity within 30 minutes when in load following mode [31, 32]. Most modern nuclear reactors can operate flexibly, including many, if not all, of the small modular reactors currently being developed [33]. Small modular reactors offer additional flexibility because individual reactors can be shut off to further reduce the output of the power plant as a whole [33]. We expect that the oil shale power plants can adjust their output at a rate of approximately 2.5 $MW_e$/min (or about 1–2%$P_{nom}$/min). A recent IRENA report shows a similar rate for coal power plants [34]. For nuclear power plants a rate in the range of 1–3% $P_{nom}$/min is also common [31]. Therefore, we included this flexibility in the production models for oil shale and nuclear power plants by decreasing their output during periods when production exceeded demand (including some potential exports from foreign demand). The rate of change was kept reasonable to make sure it corresponded to the response rates actually achievable with these types of plants. For oil shale power plants, it was assumed that the lowest possible output they could operate at was 40% of their full capacity [34]. For nuclear power plants the minimum was 20% of their full capacity [31].

It is also worth noting here that although the majority of nuclear power plants worldwide have quite large capacities, there are also multiple types of small reactors in use commercially. For instance, in India many small reactors have been built starting in the 1960s. Many of them have been domestically manufactured pressurized heavy-water reactors that have a capacity of about 200 $MW_e$ [35]. And in recent years, there has been increasing interest in small modular reactors [36, 37]. Therefore, there are vendors available that could supply a nuclear power plant with the 530 $MW_e$ capacity assumed here.

Pyrolysis gas is also used to produce between 700 and 1000 GWh of electricity in Estonia [23]. This pyrolysis gas is a byproduct of the shale oil production process, and is sometimes called semicoke gas or generator gas. In Estonia, oil shale is also used to produce a synthetic crude oil, called shale oil, via pyrolysis, and in addition to the oil a gaseous byproduct is produced. Eesti Energia feeds its pyrolysis gas to its Auvere and Estonia power plants for co-combustion [38, 39]. VKG has a separate combined heat and power plant specifically for burning pyrolysis gas, and this complex has a capacity to produce 87 MW of electricity [40, 41]. Based on data from recent years about the amount of electricity produced from pyrolysis gas [23], we estimate that there is approximately 150 $MW_e$ of capacity from pyrolysis gas in Estonia. Eesti Energia is planning to build an additional shale oil plant in the next few years [42], and based on this information we assumed a slightly larger pyrolysis gas capacity of 180 $MW_e$ in the simulations. However, because pyrolysis gas is a byproduct, it cannot reasonably be assumed that it could provide enough base power to stabilize the Estonian grid on its own. Therefore, the capacity was kept constant at 180 $MW_e$ in all the simulations in which pyrolysis gas was included.

Transmission and distribution losses were also deducted from the amount of electricity produced. These losses are about 8% in Estonia [43].

Note also that in the simulations production was not required to exactly meet demand. This is because Estonia is connected to the Nordic electricity grid (Nordpool). It has connections with Latvia and undersea cables connecting it with Finland. Through Finland and Latvia it is also connected with Skandanavia and Lithuania. Therefore, surpluses and shortfalls can be compensated for by importing or exporting electricity to the Nordpool grid, which helps keep the Estonian grid in balance. When the simulation gave extremely large surpluses or shortfalls for a particular system, this indicates that the specific system would most likely not be realistic and would not be a good option.

## Storage

For some of the configurations, energy storage was included. This was modeled by sending excess electricity to the storage system when production exceeded demand and drawing on energy in the storage system when production fell short of demand. The amount of energy transferred was limited by the power rating of the storage system. Both the power rating and the energy capacity could be adjusted to simulate different types of storage systems. Efficiencies were also taken into account both when charging and discharging. The exact algorithm used can be seen from our code: https://github.com/zmeri/electricity-sim-estonia/blob/master/storage_func/storage_base.cpp

For the purposes of this simulation, we assumed that surplus electricity was first stored before exporting.

There are a variety of potential energy storage technologies. We chose to use underground pumped hydro storage in our simulations because it is currently the most promising technology for Estonia. Pumped hydro storage is a well-established technology that currently accounts for the majority of global energy storage [44]. Because Estonia does not naturally have large changes in elevation, the height differential required could be achieved by placing the lower reservoir underground. Indeed, the company Energiasalv is attempting to construct such an underground storage facility in Northern Estonia near Paldiski [45, 46]. Eesti Energia is also planning a smaller 50 MW underground pumped storage project in Ida-Virumaa county in Northern Estonia [47]. Also, pumped hydro storage is generally estimated to be one of the cheapest forms of long-term energy storage.

Many of the other potential energy storage technologies are currently much more expensive, such as battery technologies [48, 49]. For this reason, these technologies were not considered. Additionally, many storage technologies have yet to reach a mature stage of development, which is another reason we did not consider them for this study. We did perform a few simulations using hydrogen storage for comparison; however, the cost of hydrogen storage was significantly higher than that of pumped hydro storage. In large part this seems to be due to the fact that fuel cells and electrolyzers currently cost much more than pumps and turbines with the same power rating [50–55]. Fuel cells and electrolyzers also generally have lower efficiencies than pumped hydro storage, which would also cause hydrogen storage to have a comparatively higher cost [52, 56]. Also, storing the hydrogen might be challenging because Estonia does not have suitable underground formations, such as salt caverns, so more energy intensive pressurized or cryogenic options would need to be used [57]. Therefore, we chose underground pumped hydro storage as the technology for comparisons in this study.

The round-trip efficiency for underground pumped hydro storage was taken to be about 75%, so the efficiency for a single charge or discharge was set to be 86.5% [56].

## Import and export

We assumed that it is possible to export an amount equal to 30% of the electricity demand in Estonia. This value was chosen based on data from 2018 in which net exports from Estonia were about 26% of domestic demand [58]. It is worth noting that in 2019 and 2020 production from oil shale power plants dropped, probably due to higher carbon emission prices, and currently Estonia is no longer a net exporter [28, 29, 58]. However, if economically competitive electricity production would be available in Estonia, we assume that exports similar to those in 2018 could be achieved. The electricity connections with Latvia and Finland can handle up to 2463 MW of exports and 2275 MW of imports [59], as of January 2021, so it seems exports would be limited mainly by the demand in neighboring countries and not the power rating of the transmission cables.

Any surplus electricity that could not be exported was considered to be unused potential. In practice, this surplus is often handled by curtailing electricity production, such as by rotating the vanes of wind turbines to reduce production, or by paying other grids to accept the excess power [60].

## Cost estimation

Here we describe some of the assumptions and methods used to estimate the levelized cost of electricity (LCOE) for oil shale plants with CCS and alternative technologies. The Python code and other materials used in these analyses can be found in the Github repository for this project (https://github.com/zmeri/electricity-sim-estonia/).

In our calculations we used a Monte Carlo method. In essence, this involved specifying a distribution for the parameters affecting the cost of a given technology and then sampling many times from each distribution and combining to get an overall expected range for the levelized cost of electricity. Using this Monte Carlo method allowed us to take into account the uncertainty of the parameters we used. For the simulations investigating the energy storage required to achieve balanced electricity imports and exports (subsection "Energy storage required"), the simulation was repeated 1900 times for each set of parameters for the storage system. For the simulations comparing 8 different likely scenarios (subsection "Comparison of potential systems"), 3003 repeats were performed.

For all the calculations we performed we assumed a discount rate of 9% and 2020 euros was used as the base currency. The formula we used to calculate the LCOE is the one most frequently used. It is presented in an IRENA report [5] and elsewhere in the literature, and we show it here as Eq 1.

$$\mathrm{LCOE} = \frac{\sum_{t=1}^{n} \frac{I_t + M_t + F_t}{(1+r)^t}}{\sum_{t=1}^{n} \frac{E_t}{(1+r)^t}} \tag{1}$$

In Eq 1 *LCOE* is the average lifetime levelized cost of electricity generation, $I_t$ are the investment expenses in year $t$, $M_t$ are the operation and maintenance expenses, $F_t$ are the fuel expenses, $E_t$ is the electricity generation for year $t$, $r$ is the discount rate, and $n$ is the lifetime of the system.

**Oil shale.** For the oil shale plants, we first needed to estimate the current cost of electricity. The overnight construction cost could be estimated based on the cost of the Auvere oil shale power plant, which was constructed in Estonia between 2012 and 2015. The total cost of the plant was 610 million euros and it has the capacity to produce 300 MW of electricity [61]. To calculate the overnight construction cost, the interest during construction must also be excluded [35]. The interest during construction includes both the cost of debt as well as the

amount needed to provide an acceptable rate of return to equity investors [62]. Using the formula given in Section 2.3.4 of reference [62], a construction time of 4 years, and the discount rate used throughout this study (9%), we estimated the interest during construction and subtracted that from the reported cost of the Auvere plant. We also accounted for inflation to update to 2019 euros using the Chemical Engineering Plant Cost Index [63]. Based on this calculation, we estimated the overnight construction cost of a new oil shale power plant to be about 1870 EUR/kW$_e$.

To estimate the operating expenses, we used information from the media and the literature. For instance, on July 2, 2019 Eesti Energia stopped electricity production because its production cost was higher than the market price [29]. The wholesale market price was about 30–50 EUR/MWh at that time [64]. Also, Eesti Eneriga has stated that the cost of $CO_2$ credits makes up more than half of the production cost of electricity for the oil shale power plants [28], and these credits cost about 25–27 EUR/MWh at that time because oil shale emits about 1 tonne of $CO_2$ per MWh produced [29]. Additionally, it takes approximately 1 tonne of oil shale to produce 1 MWh of electricity, and oil shale costs about 12 EUR/tonne, which sets a lower limit for the operating costs [65–67]. We also found a government document from 2005 that indicates the operating costs of the oil shale power plants was approximately 30 EUR/MWh [68]. However, these older plants had a lower efficiency of about 26–30%, and a new plant would probably have an efficiency higher than 40%. With the higher efficiency the operating costs would also be expected to decrease to about 20 EUR/MWh. Based on this information, the operating costs for oil-shale-fired power plants is probably somewhere between 15 and 25 EUR/MWh.

We also included the cost of $CO_2$ credits when calculating the LCOE. The price of $CO_2$ on the European Emissions Trading System can change relatively rapidly. For instance, between November of 2020 and May of 2021 the price of $CO_2$ increased from about 27 EUR/tonne to about 50 EUR/tonne [69]. This makes it difficult to estimate what the price might be in the future and adds uncertainty to our estimate of the LCOE for oil shale. We used an average value of 40 EUR/tonne and a wide standard deviation of 15 EUR/tonne on the distribution to account for this uncertainty in the price. When excluding the cost of $CO_2$ credits, we estimated the LCOE of an oil shale plant without CCS to be approximately 60 EUR/MWh.

The newer Auvere plant has also been designed so that it can use up to 50% biomass in its fuel mixture, and indeed, large amounts of biomass are currently used. In our analysis, though, we considered biomass and oil shale separately. There is not enough biomass in Estonia to provide the base load needed to stabilize the grid [70]. Most of the wood available for energy use is already being used, and this capacity is already included in the simulations [70]. Although Estonia has other potential biomass resources, such as manure or energy crops like grasses, the size of these resources is still smaller than Estonia's electricity demands and the cost of these resources would likely be prohibitively expensive [71–76]. Therefore, it is unlikely that biomass could replace oil shale as the base load generator in Estonia.

To estimate the cost of electricity if CCS would be incorporated into the plant, we used the full range of cost estimates given in the literature [27, 77–113]. This data is presented in another article of ours and is also available on Open Science Framework (https://osf.io/8b2dw/). We also calculated a second estimate based on next generation capture technologies that are still being developed but are expected to have lower costs. Additionally, about 25 EUR/MWh was added to the cost to account for transportation and storage because, based on literature data, we expect that this would be the cost for transportation from Estonia to the North Sea for storage [82, 114–116]. We assumed that $CO_2$ emissions would be reduced by 90%, and accordingly we reduced the cost of the $CO_2$ credits in the LCOE estimate.

It should be noted that CCS technologies are still new and under development. The only technology that has so far been applied commercially is absorption with amines, and that has

only been used at two power plants [117]. Note that the cost estimates we used here were for an nth-of-a-kind plant, which means that several such commercial plants have already been built. If a first-of-a-kind CCS plant is selected instead, then the cost can be expected to be higher.

**Solar.** Although we could obtain data on the global cost of utility scale photovoltaics [5], we wanted to perform an estimate specifically for Estonia because it has poorer than average solar resources. For example, according to the World Bank's Solar Atlas [25], solar panels in Estonia produce about 1000 kWh annually per kW of installed capacity. By comparison, in Spain that value is about 1600 kWh/kWp and in the most sunny areas in the world the solar potential can exceed 2000 kWh/kWp.

To calculate the cost of solar electricity in Estonia specifically, we took estimates of the capital and operational cost of solar panels from the IRENA report [5] and the solar potential in Estonia from the Solar Atlas [25]. Then we calculated the discounted costs and energy production over the lifetime of the solar panels (assumed to be 25 years).

**Nuclear.** Based on literature data for the cost of nuclear power plants, we estimated the LCOE for a nuclear power plant in Estonia. The construction cost is by far the largest contributor to the cost of electricity for nuclear power plants. In estimating the overnight construction cost, we focused on small modular reactors (SMR), which is the type that has most commonly been proposed in discussions in Estonia. More specifically we estimates for Generation III + SMRs. Unlike the large nuclear reactors that have traditionally been built, SMRs would have a capacity of only dozens to maybe a few hundred $MW_e$. These smaller reactors have been touted as safer and less prone to the construction delays encountered with large reactors. Additionally, their smaller size makes it more feasible to implement them into smaller electricity grids, such as in Estonia.

To estimate the overnight construction cost of small modular nuclear reactors, we relied mainly on the expert assessments published by Abdulla et al. [118] for Generation III+ SMRS, and also took into account data from other literature sources [119–121]. Based on these sources, we used 4400 EUR/$kW_e$ as the average overnight construction cost for an nth-of-a-kind plant. This is in the same range as data for existing conventional nuclear reactors [35, 62, 118, 122]. On average, literature estimates seem to show that small modular reactors have a slightly higher overnight construction cost per kW than conventional large reactors. However, there is also the potential for cost savings with small modular reactors due to aspects such as the potential for factory fabrication and shorter construction times [118].

The actual cost of a nuclear reactor can vary widely, as can also be seen from historical data [35]. The experts interviewed by Abdulla et al. [118] also gave estimates ranging from 3200 to 7100 USD/$kW_e$, which again highlights the variability in the cost. So, we used a wide distribution to account for this uncertainty. We chose to use a lognormal distribution to describe the potential range of the capital cost because the historical data seems to follow such a distribution [35]. Using a lognormal distribution also allowed us to take into account the risk that the construction can go well over budget.

The estimated value used for fuel costs (including waste disposal) was 6 EUR/MWh and for operations and management expenses 10 EUR/MWh. These estimates were taken from D'haeseleer [62] and were similar to those given elsewhere in the literature [123, 124].

Note that small modular reactors are still a new technology that is actively being developed. One 70 MW plant is already in commercial operation in Russia. It is a floating nuclear plant in the Arctic Ocean that uses two KLT-40S reactors normally used for nuclear ice-breakers [37, 125]. This highlights how some of the small modular reactors simply use smaller scale versions of existing technology. To some extent, small reactors have already been around for decades because they have been used to power naval vessels and smaller 200 MW reactors have been

built in India for decades [35]. In 2021 a 210 MW plant is supposed to come online in China, and this will use a high-temperature gas-cooled reactor [37, 125–127]. Construction has also resumed on a 25 MW reactor in Argentina, which is expected to be completed in the next year or so [37, 127]. Several more small modular reactors are also planned or under construction [37]. So, the technology is just now reaching commercial implementation and has not yet been widely adopted. If a small modular reactor is selected for Estonia that has not been built before at an industrial scale, then a higher cost should be expected than the estimates given in this article.

**Pyrolysis gas.**   Pyrolysis gas is a byproduct of the pyrolysis process used to produce shale oil from oil shale. There are several shale oil plants in Estonia that produce the gas. Pyrolysis gas has a number of cost advantages over oil shale. First, because it is a byproduct it is essentially a free fuel. Second, we expect that the overnight construction cost of a pyrolysis gas power plant would be somewhat less than an oil shale power plant: we used 1600 EUR/kW$_e$ as the estimate for our calculations. Third, less $CO_2$ is emitted per unit of electricity produced, which means CCS would cost less.

For the $CO_2$ emission factor for pyrolysis gas, we used an average value of 0.2 tonne $CO_2$/MWh. There are two different types of shale oil production technologies used in Estonia: gas generator and solid heat carrier retorts. These different technologies produce gas with different compositions and heating values. The pyrolysis gas from solid heat carrier retorts has a higher heating value and a lower $CO_2$ emission factor. Because gas from each type of retort is combusted in roughly equal proportions, we simply took an average of the emission factors of the two different types of pyrolysis gas [128].

To estimate the cost of CCS, we used the same estimates as used for oil shale power plants, but because pyrolysis gas emits less $CO_2$ the overall contribution of CCS to the LCOE is smaller for pyrolysis gas.

It should be noted that for pyrolysis gas to be available shale oil production must continue in Estonia. Although shale oil is largely exported, and so the CO2 emissions from burning it do not generally add to Estonia's reported total, it still affects the environment. In the future there may also be a desire to end shale oil production in Estonia, and this would remove pyrolysis gas as a potential resource. Currently though, shale oil production is still expected to continue.

**Biomass and hydro.**   The estimated cost of electricity production from biomass was taken from an IRENA report [129], and the average estimate was 100 EUR/MWh. For hydropower the estimated average cost used was 80 EUR/MWh [130].

Note that we also calculated the LCOE of producing electricity in a power plant similar to the one used for oil shale. This was done because biomass is currently co-fired with oil shale in some power plants in Estonia and we wanted to compare the cost of such co-firing. For this reason, in the calculation we used essentially the same parameters as for an oil shale power plant except that we used a different price for fuel and a slightly lower efficiency (as would be expected for biomass). The LCOE calculated was added to Fig 3 for comparison.

However, in all the other simulations and analysis performed in this article we used the LCOE for biomass taken from the IRENA report [129]. This was because much of the biomass in Estonia is used in combined heat and power plants and such plants have different costs.

**Storage.**   We took data and estimates from the literature to estimate the cost of underground pumped hydro storage. Energiasalv has estimated that their proposed underground storage facility in Paldiski, Estonia would take 8 years to construct and have a lifespan of 60 years, so we used those parameters for our calculations [45]. We took estimates for the capital costs of the reservoirs, pumps, turbines, and other equipment from Kapila et al. [55] and Guo et al. [50]. Estimates from Guo et al. [50] indicate that underground pumped hydro storage is

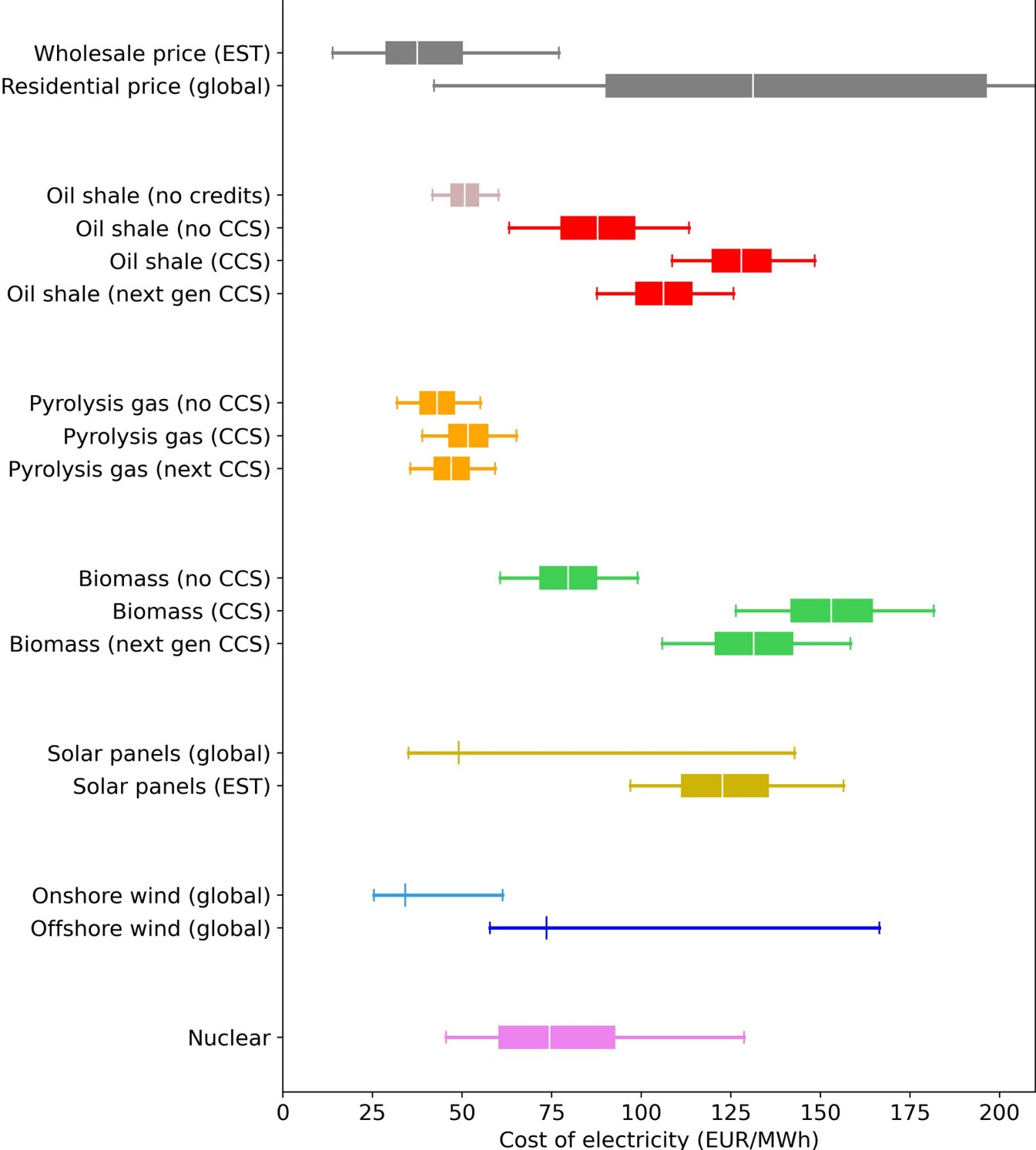

**Fig 3. Comparison of the cost of various electricity production technologies.** Global costs for solar and wind were taken from ref [5], and the average line for those is the global weighted average. The global residential price data is from [138].

roughly twice as expensive as conventional aboveground configurations, and this difference was taken into account when estimating the cost. The operational costs and scaling factors were also obtained from Kapila et al. [55]. We also took into account the cost of electricity for charging and the potential revenue gained when discharging.

**System LCOE.** Because production from solar and wind is intermittent, these energy sources incur additional costs on the entire electricity system [131]. Such costs include additional power generating capacity for when wind turbines and solar panels are not producing enough (balancing costs), additional transmission lines and more complex congestion management (grid costs), and loss of revenue due to overproduction or load reduction at dispatchable power plants (profile costs). These additional costs are not accounted for in the LCOE of solar and wind separately [132].

Therefore, to compare different potential electricity systems, we used the idea of a System LCOE, which was proposed by Ueckerdt et al. [132], to take into account these additional costs. We calculated the balancing and profile costs as part of the simulations. The balancing costs were incorporated by using the simulation data to determine how much additional production or storage capacity would be needed to achieve a balanced level of imports and exports. The profile costs were also calculated automatically in the simulation because reductions in electricity production were directly accounted for in the calculation of the cost. For the grid costs, we used literature estimates of the specific cost of grid reinforcements for wind power [131]. We multiplied this specific cost by the capacity of wind and solar installations and added the respective value to the capital cost of each.

## Job creation

We also briefly compared the various low-emission technologies based on other environmental and social factors. One of the metrics we used for comparison was the estimated number of jobs created. We calculated the number of permanent jobs created using information in the literature. For this analysis we only considered permanent jobs directly created at the energy companies producing the electricity. Additional jobs are created during construction, but because these jobs are only temporary, we left them out of the calculation. Also, many indirect jobs are also created in the surrounding economy, for example in companies that serve the energy company and in local businesses where employees of the company spend their money. However, these indirect jobs created were not included in this calculation because they are more difficult to quantify and doing so would require a separate detailed study.

For oil shale power plants with CCS, we estimated the number of jobs from mining and operations based on 2020 employment data for Enefit Power (formerly Enefit Energiatootmine AS and Enefit Kaevandused AS), which is the oil shale power plant operator in Estonia (see Table 6 in [133]). Enefit Power also produces shale oil, so we split the total number of employees between the electricity and shale oil businesses based on the percentage of the oil shale used for each product. Data for Enefit's 2019 oil shale mining and shale oil production was obtained from the yearbook for the Estonian Oil Shale Industry [134]. The amount of oil shale used specifically to produce shale oil was then estimated by assuming that 0.2 tons of shale oil is produced from 1 ton of oil shale. Based on this calculation we estimate that about 70% of the oil shale is used for electricity production. Assuming that company employees are split between the two products by a similar proportion, we calculated that electricity production employed about 1050 people at Enefit Power in 2020. Given that the total installed capacity of Enefit Power's oil shale power plants was 1490 MW, this gives a specific ratio of 0.71 jobs per MW of capacity. Including a CCS plant would also add more jobs, and based on data from the Petra Nova plant in Texas, USA, a CCS plant would create an additional 0.071 jobs per MW

[135]. Adding these ratios together and multiplying by the 600 MW of oil shale plant capacity assumed in our scenario used for comparison gives approximately 470 jobs created.

For the nuclear plant, we directly used estimates provided by Hõrak [119] in a study commissioned by Fermi Energia. They estimated that a 300 MW small modular reactor would require 170 employees. For our comparison scenario, 530 MW of nuclear power was assumed, which would create about 300 jobs.

For the energy storage technology, the jobs created by additional wind turbines and solar panels must also be included. This is because energy storage needs to be coupled with something that produces electricity, so to appropriately compare with oil shale and nuclear plants (which produce their own electricity), the additional wind turbines needed to produce the same amount of electricity were also included. For the pumped hydro storage, a ratio of 0.25 jobs/MW was calculated based on the number of people employed in operations and management at hydropower plants in the United States [136]. For the 1300 MW of hydro storage in our comparison scenario, this equates to about 330 jobs. To this we added the operations and management jobs created by wind turbines. For wind turbines, 0.36 jobs are created per MW of capacity, based on an average of literature estimates [137]. For our scenario being compared, 1655 MW of additional wind turbines would be needed to provide the same amount of electricity as the oil shale or nuclear plants. This means that about 600 jobs would be created by those wind turbines.

### Sensitivity analysis calculations

To estimate the sensitivity of the results to variations in individual parameters, we ran some simulations where only one parameter at a time was allowed to vary. The remaining parameters were assigned their average values. For each parameter, the simulation was run 200 times and by calculating the variation of the average system cost and the net surplus the effect of each parameter could be estimated. Parameters related to hydro and solar power were not included in the analysis because the capacities for these two energy sources were much smaller and adjusting their parameters did not have a significant impact on the simulation results. The code for this sensitivity analysis is also included in the Github repository for this project.

## Results and discussion

### Cost of electricity production alone

As a starting point, we compared the cost of various electricity production methods. The cost of different production technologies is generally compared using the LCOE (see Eq 1) [5].

The comparison is shown in Fig 3. The wholesale market price of electricity in Estonia (the Nordpool next day price from January 1, 2014 to September 17, 2021) [58], as well as data on the residential price of electricity in different countries, have been added for reference [138].

One of the first observations is that the current cost of electricity for oil shale plants is already above the wholesale market price in Estonia and is in the range of residential electricity prices. Note that we estimated the current cost based on literature data and publicly available information since Eesti Energia does not disclose its exact production cost (see the Methodology section for details on all the calculations). However, our estimate seems to be in the correct range given that within the past few years the Eesti Energia oil shale power plants have even temporarily shut down because the price of electricity was lower than their costs [29]. A large part of their costs is due to the $CO_2$ credits that must be purchased by the plants to offset their emissions under the EU emissions trading system, and the LCOE for oil shale power plants also includes the cost of those credits. When excluding those credits, we estimate the LCOE of an oil shale plant without CCS to be approximately 50 EUR/MWh. This shows that the current

climate strategy of the EU is putting pressure on oil shale power plants to reduce their emissions, as is intended.

Although CCS could help oil shale power plants avoid the cost of those $CO_2$ credits, implementing CCS will still lead to a significant increase in the cost, which may be one of the reasons it has not yet been implemented. Here we collected a wide range of literature estimates of the cost of carbon capture and storage to estimate the additional cost of CCS (data available on Open Science Framework at https://osf.io/8b2dw/). As seen in Fig 3, adding CCS significantly increases the cost of electricity for oil shale. Indeed, this is often cited as a reason not to invest in CCS [139]. For instance, in 2015 the UK government canceled a competition that would have provided 1 billion GBP for large scale CCS projects [140]. One of the major justifications was that CCS would cost too much money [141]. The next generation of CCS technologies is expected to have lower costs, as shown in Fig 3, but more development will be required and there is the chance that those lower estimates will not be realized at the industrial scale.

But, the lack of strong support for CCS is about more than simply its cost. In many cases the decision has been to support renewable technologies instead of cleaning up existing fossil fuel operations. Large-scale commercial wind and solar installations have become a reality, and along with the increased adoption have come significant decreases in the cost of these renewables [5]. By contrast, based on the literature data that we have collected, the cost estimates for CCS have not changed significantly in the past two decades (see the literature data we compiled: https://osf.io/na5gy/). This is likely because any new commercial projects have still used the same absorption technology. So far no one has been willing to put up the funding to construct an industrial scale plant using a new CCS technology.

The one situation in Estonia where CCS could be economically favorable is with combustion of pyrolysis gas. Pyrolysis gas is a byproduct of the production of shale oil. Shale oil is a synthetic crude oil that is also produced from oil shale via pyrolysis, and a mixture of gaseous compounds is also formed during the process [128, 142]. Because pyrolysis gas forms less $CO_2$ per unit of energy, the cost of CCS per MWh of electricity is significantly lower (see Fig 3). Additionally, because pyrolysis gas is a byproduct it is essentially a free fuel for the oil shale companies. Between 700 and 1000 GWh of electricity is already produced annually from pyrolysis gas in Estonia [23], and based on its lower cost and carbon emission factor, it could be advantageous to continue using it and implement CCS for it.

Wind and solar power have also become quite affordable as a result of consistent cost reductions over the past decade. Now the LCOE for wind and solar technologies has fallen to the level where it can directly compete with current fossil fuel plants [5]. Recent estimates of the cost of solar and wind from an IRENA report [5] are shown in Fig 3 and highlight how the global average cost for solar and onshore wind is even competitive with the current oil shale power plants without CCS. Granted, solar cells perform worse than average in Estonia due to the poor solar potential [25], so when we recalculated the LCOE for solar in Estonia specifically it is much higher. And if the choice is between implementing CCS or renewables, it does not appear that CCS has a cost advantage. Offshore wind can provide electricity at costs lower than an oil shale plant with CCS and solar has roughly the same LCOE. Given the negative public image that fossil fuels have, it could be difficult to gain support for CCS over renewables, especially if no cost benefit exists.

One important advantage that an oil shale plant has over wind and solar is stability. Renewable energy sources that are intermittent, such as solar and wind, need to be balanced with technologies that can stabilize the grid and reliably meet demand even when the sun is not shining and the wind is not blowing [143]. Additionally, the grid needs to have some way of stabilizing important parameters of the power grid, such as the frequency [143]. Currently, this is accomplished by large plants that produce a steady, stable output, which generally

means coal, natural gas, nuclear, or hydroelectric power plants. In Estonia specifically, the oil shale power plants can provide this reliable base load needed by the power grid. If Estonia were to shut down its oil shale power plants, it would need either a nuclear power plant or utility-scale energy storage system to meet the load.

A nuclear power plant would likely cost less than implementing CCS, as illustrated in Fig 3. We estimated the LCOE of a nuclear power plant based on literature data. We used data on the construction costs of small modular reactors, since these are the reactors currently being considered in Estonia, but using data for conventional large reactors gives a similar LCOE. Fermi Energia is also actively promoting the construction of a nuclear power plant in Estonia. For comparison, we used the budget they have published to calculate the LCOE of their proposal [119]. Based on our calculations, the LCOE for the Fermi Energia nuclear plant would be 78 EUR/MWh, which is quite similar to the median LCOE we calculated based on literature data.

The LCOE is significantly less than that of an oil shale plant with CCS. Based on our cost estimates, there is a 94% chance that nuclear would have a lower LCOE than an oil shale plant with current CCS technologies. Even if a next generation CCS technology would be used, which are expected to be cheaper, but require more development, there is an 85% chance nuclear would be cheaper.

Other sources have also indicated that a nuclear plant could be preferable to a fossil fuel plant with CCS. For instance, the Intergovernmental Panel on Climate Change also estimated that a nuclear plant would cost less than a coal plant with CCS (see Fig TS.19 in [144]). The IEA, in their report "Projected Costs of Generating Electricity 2020", also estimated that using CCS with coal or lignite power plants would cost about 50 USD/MWh more than nuclear [145]. And in a recent report, the Stockholm Environment Institute recommended transitioning Estonia from oil shale to electricity based on wind turbines and a nuclear power plant [146].

One final renewable resource is biomass, but it cannot be considered a viable replacement for the oil shale power plants. The reason is that there is not enough biomass to replace them. Although Estonia does have significant forest resources, more than half is used for non-energy purposes [70]. Of the approximately 5.5 million m$^3$ that is currently used for energy production, about 2.3 million m$^3$ is burned directly in homes for heating. The remaining 3.2 million m$^3$ from the forest, along with about 2 million m$^3$ of waste from the wood industry, is already used for energy production, mainly in combined heat and power plants or for producing pellets [24, 70]. Therefore, to add the approximately 600 MW$_e$ of capacity needed to provide a stable base of dispatchable power, new biomass sources would be needed. The only other biomass resources with significant potential in Estonia are manure (for biogas production) and energy crops (grasses). However, the estimated potential for these sources is, respectively, 4.5 and 21 PJ [71–75]. For comparison, Estonia uses about 8.4 TWh of electricity per year [76]. Given the efficiency of generating electricity from thermal energy, which is generally 30–40% for biomass combustion, only about 2.5 TWh in new generation could potentially be obtained from biomass, which would be about 285 MW$_e$ of capacity. Thus, even if all the manure could be utilized and grasses could be grown as energy crops, they could not provide enough base capacity on their own. Additionally, collecting and utilizing large amounts of manure and grasses would be logistically challenging and likely expensive [129, 147].

Furthermore, the LCOE for production from biomass is higher than that of oil shale (excluding $CO_2$ credits). This means that cofiring biomass with oil shale in a plant with CCS would cost even more than just using oil shale with CCS. Although biomass combined with CCS is often touted as a way to achieve negative life cycle emissions of CO2, it would be more costly. If in the future some sort of credit was provided to promote CCS with biomass, then it

might make economic sense, but currently no such incentive is available in Estonia. And, it should be noted that the LCOE shown in Fig 3 is for biomass used only for electricity production. The LCOE for a combined heat and power plant would be different and potentially higher [129]. Due to the limited resources and higher cost, only the existing biomass capacity was included in further simulations and analyses.

## Energy storage required

The cost of electricity production alone is only one piece of the picture. Because wind power is non-dispatchable, energy storage also needs to be included in the grid if wind power is to make up a majority of the electricity production in Estonia.

We first investigated how much electricity storage would be needed to achieve a certain net surplus/deficit in Estonia. For these simulations, only renewable resources were included: so wind turbines (3800 MW) and the solar, biomass, and hydropower capacity already available. 3800 MW was selected as the wind turbine capacity because, at this scale, the raw amount of electricity produced would be approximately equal to the demand in Estonia, which means that any shortfalls are mostly caused by the intermittency of wind.

We chose to model the energy storage as an underground pumped hydro system. We selected this technology because pumped hydro storage is mature and is currently one of the cheapest forms of energy storage [48, 49]. Also, two Estonian companies, Energiasalv and Eesti Energia, are currently attempting to construct underground pumped hydro storage facilities in Estonia [45–47].

Fig 4 shows how an increasing amount of electricity storage affects the net electricity surplus/deficit, the amount of overproduction, and the total amount imported for a year with such a system. For reference, Estonia's electricity consumption for all of 2016 was about 8380 GWh [76]. The amount overproduced is the electricity that probably could not be used domestically or exported. In reality curtailments would likely be used to require electricity producers to limit their production. Overproduction is a common problem with electricity grids that have high shares of non-dispatchable power, such as solar and wind [60].

From Fig 4 we can conclude that at a storage capacity of about 100 GWh (with 3800 MW wind turbines) Estonia would be close to balancing electricity imports and exports. To put this in perspective, 100 GWh would be enough to completely cover Estonia's average electricity demand for about 4 days. However, curtailments on the order of 5–10% would probably still be necessary to handle times when wind turbine production exceeds the storage capacity. That is, some of the wind turbines may need to be shut off at times to avoid producing more electricity than can be used, stored, or exported. The storage capacity would need to be at least an order of magnitude larger to avoid curtailments all together. Such a large storage capacity, however, would cost significantly more and probably would not make economic sense. On the other hand, a smaller storage capacity could be used if policymakers determine that a higher amount of imports and curtailments is acceptable. Also, Fig 4 indicates that storage systems smaller than 1 GWh would have essentially no effect on the net electricity deficit.

The amount of storage needed would decrease if there was a larger capacity of wind turbines. For instance, with 4000 MW of wind turbines, only about 30 GWh of storage would be needed to balance imports and exports. This is because overall more electricity would be produced and provided to the system, reducing the need for storage. However, overproduction of electricity would also increase, leading to a higher level of curtailments. Curtailments increase the levelized cost of electricity from wind turbines because the same capital costs must be spread over a smaller volume of production. Therefore, increasing the number of wind turbines to reduce the need for storage would probably not reduce the cost of the overall system.

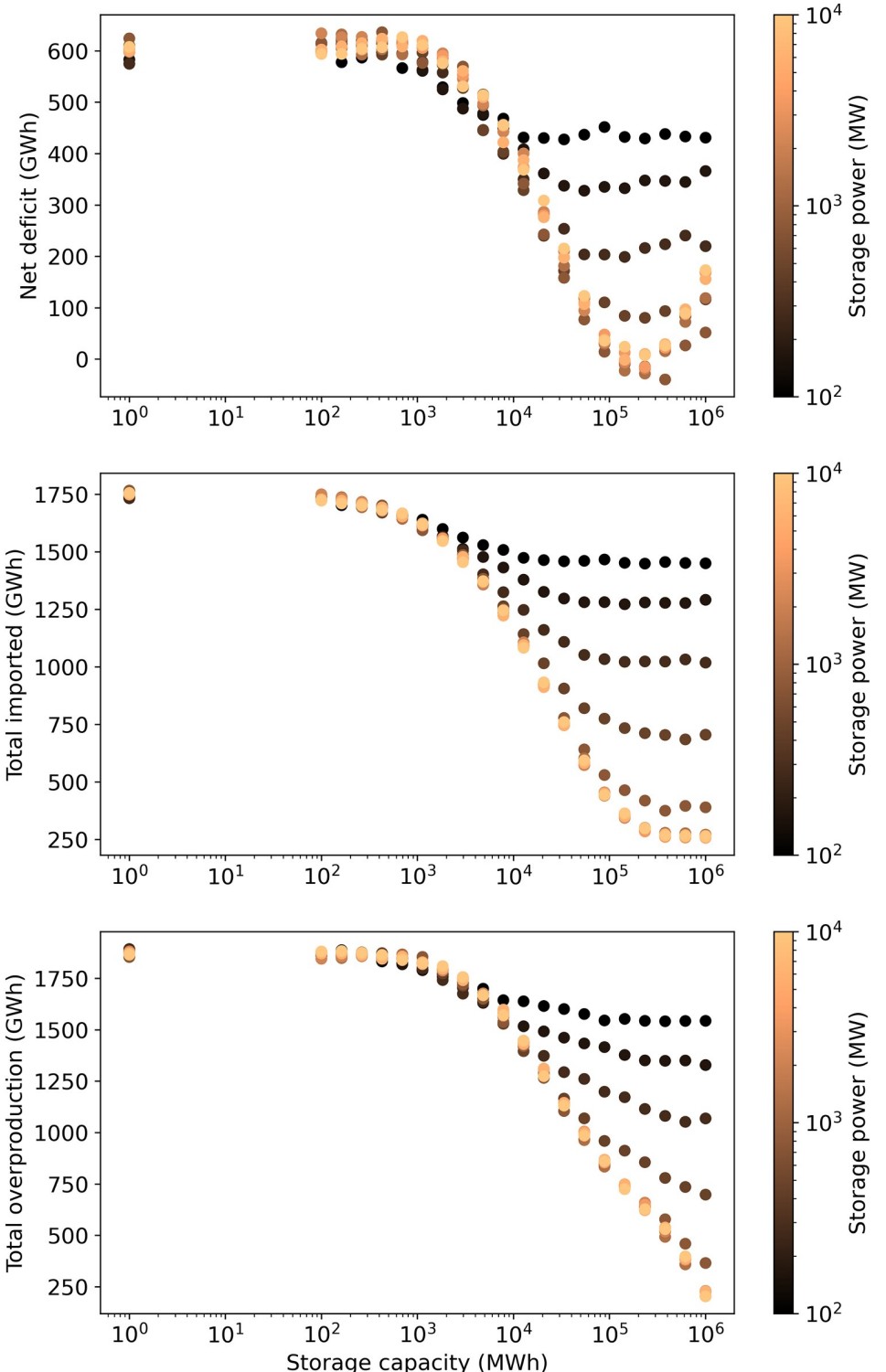

**Fig 4. Estimated effect of storage capacity and power on Estonia's annual electricity balance.**

If there is less than 3800 MW of wind turbines, it is unlikely that enough electricity could be produced to meet Estonia's demand from wind turbines (along with the existing biomass and solar capacity).

We also investigated how the power rating of the storage system affects the amount of energy lost. As seen from Fig 4, if the storage power is too low, then increasing the storage capacity has essentially no impact on the net electricity deficit. In other words, the storage power must be large enough to make use of a larger storage capacity.

It is also worth noting that at a certain point, increasing the storage capacity and storage power has basically no effect on the net deficit. For the given simulation, this occurred at about 100 GWh of storage and 800 MW of storage power. Above those levels, the system was instead limited by the installed wind turbine capacity, which was set at 3800 MW for this analysis. One potential way to use such excess capacity would be to import and store electricity from other countries. We did not investigate such scenarios in this study, though, because the more pressing question is how Estonia can still meet domestic demand when switching to a low-emission electricity system. Therefore, we assumed that the smallest viable storage system would be selected to reduce the overall cost of the storage in the electricity system.

## Comparison of potential systems

What we really want to know is the cost and performance of the system as a whole when taking into account the natural fluctuations in the grid and the need for energy storage. Although wind and solar power have competitive production costs, they are also non-dispatchable and must be coupled with another stable power supply to compensate for the fluctuations.

We selected 10 scenarios for comparison (see Table 1). To determine how much storage to include in each scenario, we used the results of the earlier analysis to estimate how much energy storage might be needed. And again, underground pumped hydro was chosen as the storage technology. For each scenario, a total capacity was selected that would produce about enough electricity to cover demand in Estonia, so the differences in the net surplus are mainly due to mismatches between supply and demand and not a lack of capacity. Only wind turbines were included as the additional renewable source because solar power in Estonia has a higher LCOE (see Fig 3). Additionally, even in the "Oil shale with CCS" and "Nuclear" scenarios 400 MW of wind turbines was included because there is already about that amount of wind capacity installed in Estonia (315 MW, as of 2019 [22]).

**Table 1. Overview of the scenarios compared**[*].

| Scenario | Wind turbines (MW) | Oil shale or nuclear (MW) | Pyrolysis gas (MW) | Storage capacity (GWh) | Storage power (MW) |
|---|---|---|---|---|---|
| Renewables only | 3800 | 0 | 0 | 0 | 0 |
| Renewables—5 GWh storage | 3800 | 0 | 0 | 5 | 500 |
| Renewables—20 GWh storage | 3800 | 0 | 0 | 20 | 800 |
| Renewables—100 GWh storage | 3800 | 0 | 0 | 100 | 800 |
| Oil shale with CCS | 400 | 915 | 180 | 0 | 0 |
| Nuclear | 400 | 970 | 0 | 0 | 0 |
| Renewables with only oil shale | 1900 | 600 | 0 | 0 | 0 |
| Renewables with oil shale and pyrolysis gas | 1900 | 415 | 180 | 0 | 0 |
| Renewables with nuclear | 1900 | 530 | 0 | 0 | 0 |
| Renewables, nuclear, and pyrolysis gas | 1900 | 365 | 180 | 0 | 0 |

[*] All scenarios also include the current electricity production from solar, hydro, biomass, and waste in Estonia (about 200 MW of capacity).

The results of the simulations for these scenarios are shown in Fig 5, and the raw data from the simulations is available on Open Science Framework (https://osf.io/8b2dw/). As can be seen, about 100 GWh of electricity storage would be needed to provide comparable stability to using an oil shale or nuclear power plant. With less storage there would be an electricity deficit and more electricity would need to be imported to make up the difference. The surplus duration curves shown in Fig 6 also show how having less storage leads to many more days in which a deficit would occur.

For both the scenarios with 100 GWh of storage and those with a thermal power plant, the amount of imported electricity would be approximately in balance with the amount exported, on average. However, systems with electricity storage would cost significantly more, even when taking into account potential revenue from electricity arbitrage. Using 100 GWh of electricity storage to stabilize the grid would, by our estimates, cost about 40% more than using oil shale and pyrolysis gas power plants with CCS and about 50% more than a nuclear power plant. This finding is consistent with the results from Zappa et al. [148] in which they estimated that a 100% renewable system in Europe would cost about 30% more than systems with nuclear or CCS technologies. Pleßmann et al. [149] estimated that the global average cost of electricity for a 100% renewable grid would be 142 EUR/MWh, which is reasonably close to the mean of 152 EUR/MWh calculated here. However, Pleßmann et al. seem to have underestimated the cost of the power to gas technology used in their model. Pleßmann et al. assumed the capital costs would be 940 EUR/kW, but other sources show that even the electrolyzer alone would cost more than that [52, 150–153]. In summary, using only electricity storage and no thermal power plants would most likely lead to significantly higher electricity prices.

We also performed some calculations using hydrogen storage instead of underground pumped hydro, but hydrogen storage was significantly more expensive. This seems to be due to the higher cost of fuel cells and electrolyzers compared to the turbines and pumps used in a pumped hydro system [52, 55]. The lower efficiency of hydrogen energy conversion might also be a contributing factor [52, 56], and large scale hydrogen storage may be more challenging in Estonia due to a lack of suitable geological formations [57]. Battery technologies, including flow batteries, currently cost significantly more than either of these options [44, 48, 49].

Although it would be quite expensive if electricity were provided almost solely by wind power, having some amount of renewables actually should help lower the cost. This is shown by Fig 7, which shows simulation results for how the cost would change as power plant capacity is replaced by wind turbines. There is a drop in the cost for wind capacities up to about 400 MW because this is roughly the potential for onshore wind turbines in Estonia (Meeliste et al. [146] estimated 500 MW as the maximum). As of 2019, there was already 315 MW of installed onshore wind turbines in Estonia [22]. Onshore wind is significantly cheaper than the other sources of low-emission electricity in Estonia, so adding this helps reduce the levelized cost. Offshore wind turbines, although more expensive than onshore ones, are still expected to be less expensive than oil shale and probably will be cheaper than nuclear if the cost of wind power continues to decrease (see Fig 3) [5]. The implication is that some amount of renewable electricity generation should be in the mix in Estonia because wind turbines (especially onshore wind turbines) are most likely the cheapest form of low-emission energy.

As the percentage of renewables in the grid increases, the net surplus of electricity decreases and the overall cost of electricity increases. As shown in Fig 7, for wind capacities up to about 50–60% of Estonia's demand, thermal power plants could still stabilize the grid enough to make Estonia a net exporter. However, at higher levels of wind penetration the power plants could not smooth the variations on their own, and additional energy storage or peaker plants (such as natural gas plants) would be needed. Also, at higher capacities, wind turbines would need to be built offshore or solar panels would need to be used, both of which are more

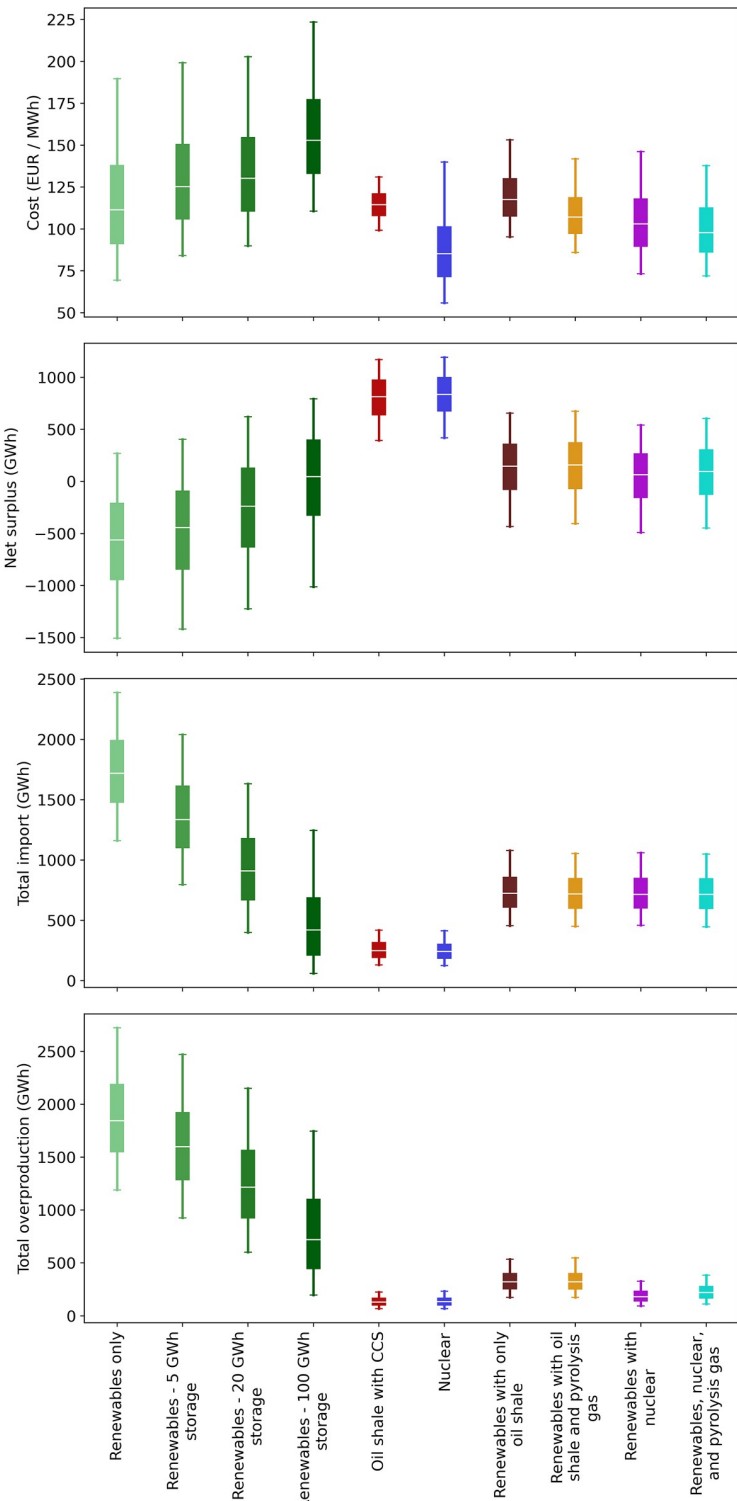

**Fig 5. Comparison of different potential energy production scenarios by the levelized cost of electricity and their effect on Estonia's electricity balance.**

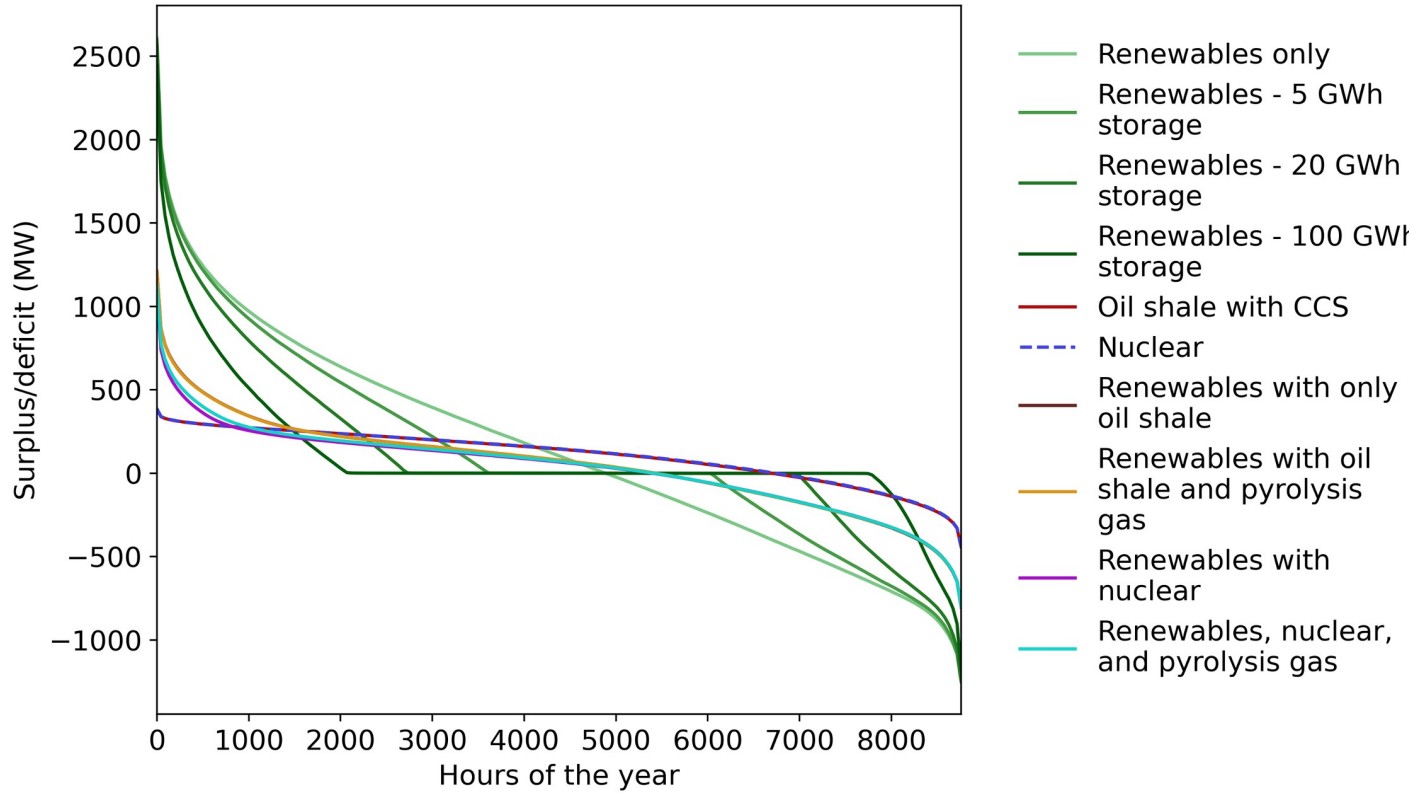

**Fig 6. Surplus duration curves for the scenarios simulated.**

expensive. These factors would increase the overall cost of electricity as more wind and solar power is added to the Estonian grid. This is why the cost of a system with wind turbines and nuclear power would increase as the proportion of wind power increases (see Fig 7).

For oil shale plants with CCS, offshore wind turbines actually have a lower cost, so initially adding more wind turbines decreases the overall system LCOE. Additionally, as the electricity generating capacity of the oil shale industry would decrease, pyrolysis gas would account for a larger share of the capacity. Since pyrolysis gas is cheaper, this would decrease the overall cost of a system based on oil shale. However, at about 2000 MW of wind capacity, the cost levels off and even slightly increases for the scenario with oil shale and pyrolysis gas power plants. This is because a high penetration of renewables starts influencing the stability of the electricity supply, which increases the system LCOE due to potential overproduction and the resulting loss in electricity production. If the wind turbines or power plants have to curtail their production, then they must spread their fixed costs over a smaller amount of electricity sold, which increases the levelized cost of a unit of electricity. The same effect also occurs with the power plants if they need to be operated flexibly to balance the intermittent wind supply. This can again be seen from Fig 5. The "Renewables with oil shale and pyrolysis gas" and "Renewables with nuclear" scenarios have a larger share of wind power (roughly 50% of the generated electricity would be from wind turbines), and consequentially the "Renewables with nuclear" scenario has a higher cost compared to the "Nuclear" scenario. The cost in the "Renewables with oil shale and pyrolysis gas" scenario is also lower than the "Oil shale with CCS" scenario. When wind power accounts for more than about 50% of production, then the amount of potential

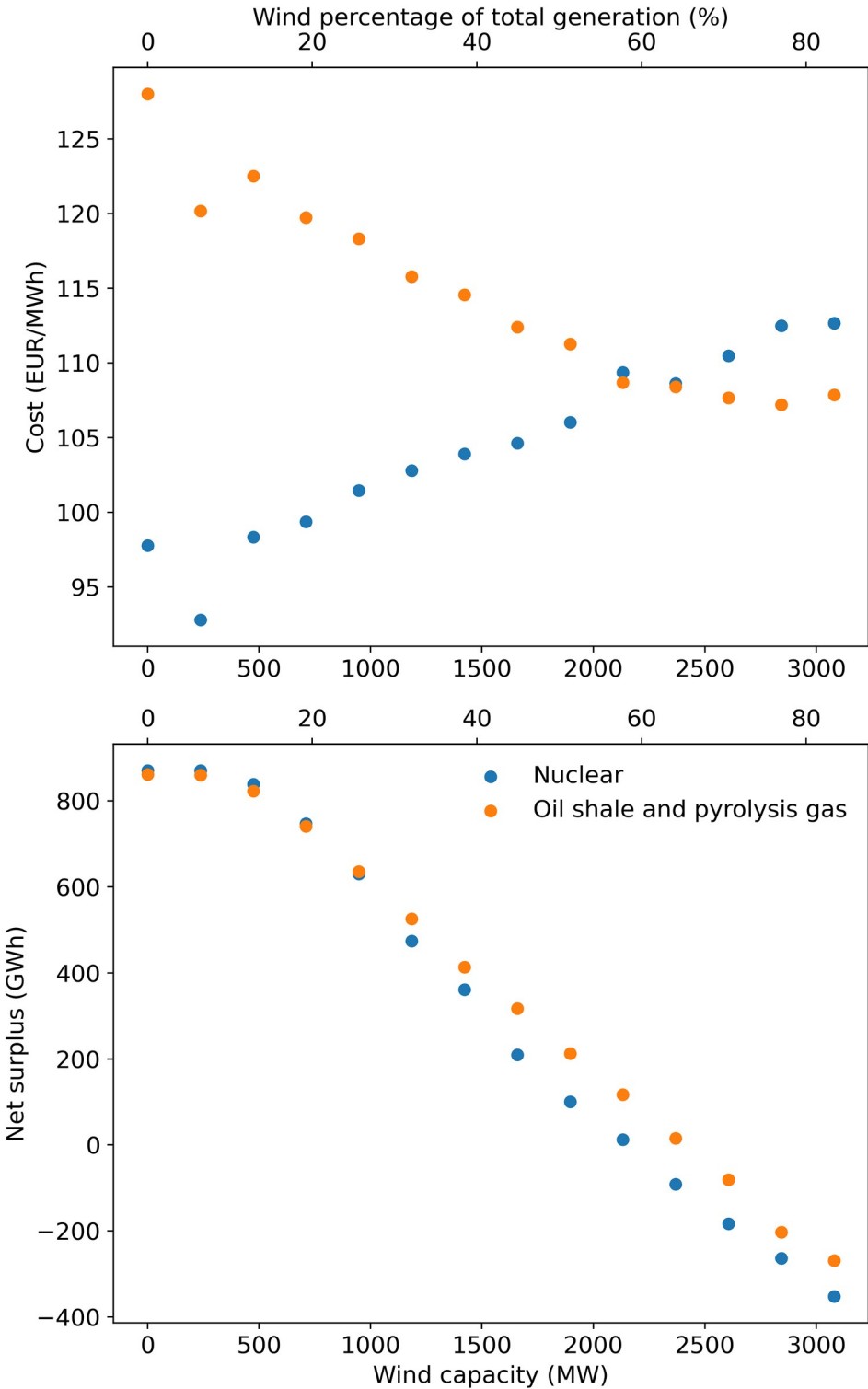

**Fig 7. How increasing the wind turbine capacity would affect the levelized cost of electricity and surplus for the electricity system as a whole.**

overproduction starts to increase exponentially and Estonia would be a net importer of electricity (at the capacities used for these simulations).

Stable, dispatchable power could be provided by either oil shale power plants or a nuclear plant, but a nuclear plant is expected to cost less. To compare the two strategies, we included a scenario where all remaining oil shale plants are replaced by 530 MW of nuclear power ("Renewables with nuclear"). Nuclear has a higher capacity factor (about 93%) [30], so this would provide the same amount of electricity as the 600 MW of oil shale power plants in the "Renewables with only oil shale" scenario. By comparing these two scenarios we can see that nuclear would likely offer two advantages. First, nuclear power would most likely be cheaper than that from an oil shale power plant with CCS (see Fig 5). Indeed, the "Renewables with nuclear" scenario does have a lower estimated electricity cost than the comparable oil shale scenario.

Producing electricity from pyrolysis gas helps the oil shale industry to lower its costs because pyrolysis gas is a byproduct and emits less $CO_2$ per unit of electricity. The oil shale industry already produces electricity from pyrolysis gas, and continuing to do so would reduce the amount of oil shale or nuclear needed and help reduce the overall cost of electricity, as shown by the scenarios that include pyrolysis gas in Fig 5. However, there is not enough pyrolysis gas available to completely provide the capacity necessary for stabilizing the grid, so an oil shale or nuclear power plant would still be needed, even when utilizing pyrolysis gas.

A nuclear power plant would also likely have the advantage of requiring somewhat smaller curtailments. This is because nuclear power plants can in general be operated more flexibly than the current oil shale power plants. For instance, nuclear power plants can generally be operated at as low as 20% of full capacity [31–33], but combustion boilers, like those in oil shale plants, generally have a lower limit in the range of 30–60%, although specialized modifications to the boiler can reduce this [34]. The effect of the better flexibility of a nuclear plant was seen in the simulations because the estimated curtailments (overproduction) were lower for the "Renewables with nuclear" scenario. Based on our simulations, and the data and assumptions behind them, a nuclear power plant would likely provide better performance than a comparable oil shale power plant.

## Sensitivity analysis

To estimate how much various parameters affected the results of the simulations, we performed a simple sensitivity analysis. We ran simulations for the three main scenarios: "Renewables—100 GWh storage", "Renewables with oil shale and pyrolysis gas", and "Renewables, nuclear, and pyrolysis gas", but this time only one parameter at a time was allowed to vary. The remaining parameters were assigned a single mean value. The resulting variation in the average system cost and net surplus are shown in Fig 8 and give an estimate of how variations in each parameter affect the simulation results.

The results in Fig 8 show that the cost of offshore wind power is the biggest source of variability in the average system cost for these scenarios. One reason is that in these scenarios wind power makes up a majority of the production capacity, and therefore, changes in the cost of wind power have a proportionally larger effect on the overall cost of electricity in the grid. A second reason is simply that there is large variability in the cost of different wind projects [5]. So, uncertainty around the cost of offshore wind also affects the uncertainty of the simulation results. Because the cost of wind and solar have fallen in recent years, this means that a lower cost might be achieved than the simulation results shown here if that trend continues [5].

Changes in the construction/capital costs of storage and nuclear also have a meaningful effect on the system LCOE of electricity. This is expected for nuclear since construction costs

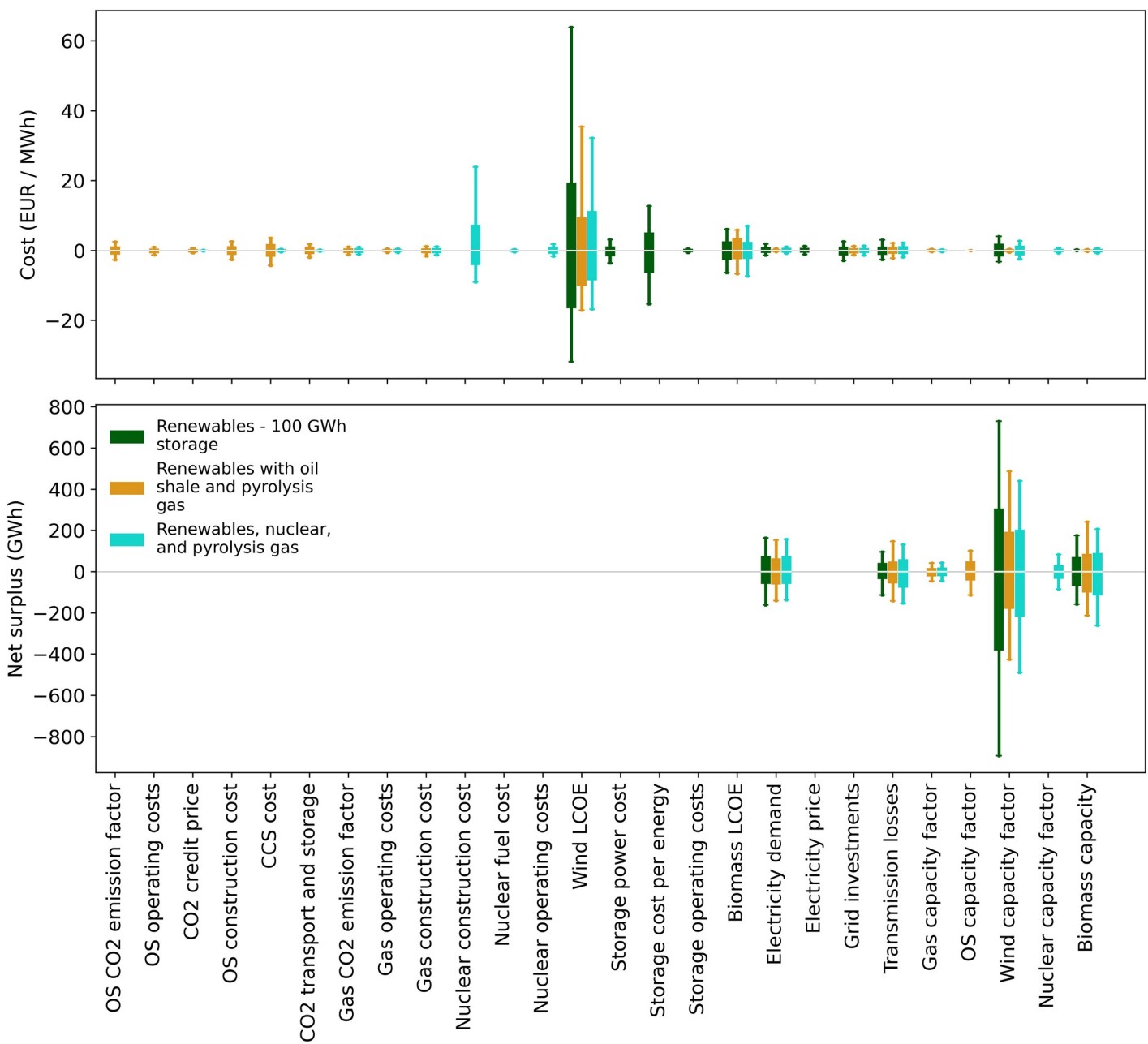

**Fig 8. Sensitivity analysis showing how the variability of individual parameters impacts the cost and net surplus of the electricity grid.**

are the largest component of the LCOE of nuclear [35, 62]. For energy storage, it seems the cost of storage per unit of energy has more of an impact than the cost of the energy conversion equipment (i.e. the cost per unit of power). For the underground pumped hydro technology specifically, this means variations in the cost of constructing the underground reservoirs have a bigger impact than variations in the cost of pumps and turbines used to move water between the lower and upper reservoirs. For oil shale and pyrolysis gas, there does not appear to be a single dominate factor that leads to variation in the system cost of electricity.

**Table 2. Comparison of the environmental impact, safety, and job creation of low-emission electricity technologies.**

|  | Oil shale plant with CCS | Nuclear plant | Pumped hydro + wind | References |
|---|---|---|---|---|
| Capacity | 600 MW | 530 MW | Storage: 1300 MW, 100 GWh<br>Wind: 1655 MW |  |
| Greenhouse gas emissions (gCO$_2$eq/kWh) | 150 | 12 | 30 | [144, 154] |
| Death rate (deaths/TWh) | 30 or 2* | 0.07 | 0.05 | [155–157] |
| Direct jobs | 470 | 300 | 930 | [119, 133–137] |

* For oxyfuel combustion there is expected to be a significant reduction in particulate matter (PM) emissions, which would lead to a lower death rate. However, for absorption processes the average expected PM emissions are roughly the same as without capture [158].

In terms of the balance between production and demand, the capacity factor of the wind turbines is the most significant factor that can lead to variations in the net surplus of electricity, at least for these scenarios. Again, reasons behind this include both the large capacity of wind turbines in the scenarios and the higher variability in production from intermittent sources like the wind. The size of the biomass capacity in Estonia also would affect the net surplus of electricity, as would variations in electricity demand and the cost of grid investments needed for transitioning to using more wind power.

## Additional aspects

Other aspects of the energy system also need to be considered when deciding which to implement. The most significant other aspects are the environmental impact, safety, and the socio-economic impact of the different systems. Although these aspects are not the focus of this article, we give a brief overview of some literature and data on these aspects to give a more well-rounded comparison of the technologies. A summary of some key metrics related to these aspects is given in Table 2 for the three potential technologies that could provide stability to the Estonian grid.

One significant factor is the environmental impact of each scenario. All of these systems would have much lower lifetime carbon dioxide emissions than the current oil shale power plants, which is positive. However, an oil shale power plant with CCS will still have much higher lifecycle emissions than wind turbines or a nuclear power plant [144]. CCS generally only captures about 90% of the carbon dioxide in the flue gas because capturing close to 100% usually increases the cost of CCS significantly [79, 82]. So, even with CCS, an oil shale power plant would emit about 100–200 gCO$_2$eq/kWh, compared with a median emission rate of about 12 gCO$_2$eq/kWh for a nuclear power plant or wind turbines [144]. Emissions of other pollutants, such as SOx, NOx, and particulate matter, are also a concern with oil shale power plants [159]. For a nuclear plant the spent fuel also needs to be handled, but this is already routinely done at nuclear plants and actually is only a small portion of the overall LCOE [62, 123, 124].

Safety is another important factor to consider. Fossil fuel sources have the highest death rates, much higher than nuclear, wind, or solar [62, 155–157]. Even when including estimated deaths from nuclear accidents at Chernobyl and Fukushima, as done in Table 2, the death rate is orders of magnitude higher for fossil fuels. The main cause of death from fossil fuels is air pollution. However, with nuclear power plants, the discussion around safety has become quite politicized and the actual risk level may be less important than the public's perception of the risk.

In addition to the safety risks of fossil fuels, job creation is also an important socioeconomic factor to consider. The complete impact of a project on jobs in a region can be difficult to

assess because in addition to direct jobs at a specific facility, other jobs in the surrounding community can be benefited, or harmed, by a project. For simplification, here we estimated only the direct jobs that would be created by the facility providing dispatchable electricity. We did not include jobs needed during construction because these would only be created temporarily during the construction phase and would require a more complicated analysis to assess their levelized impact. The estimated number of direct jobs created is shown in Table 2. The estimates are based on information in the literature [119, 133–137]. For the electricity storage system, we also included the jobs created by the additional 1655 MW of wind turbines that would be needed to provide electricity for the storage system. For details on calculations we made, see the "Job creation" subsection. The data show that each of the systems would provide a significant number of jobs. Using energy storage would probably create the most jobs due to the increased amount of renewables that could be installed. Wind turbines create a larger number of jobs per unit of electricity than either fossil fuel or nuclear plants, even when excluding construction and considering only the jobs required for operations and management [137, 160]. However, if switching, the current workers in the oil shale industry might not necessarily be able to get those jobs created by other energy technologies. Retraining or relocation might be needed. Additionally, the oil shale industry is the largest employer in Ida-Virumaa county in Estonia. If energy sector jobs move from Ida-Virumaa to another place in Estonia, then active measures would need to be taken to compensate for the loss and to help develop the local economy there [133].

Again, here we just give a brief overview of these additional aspects that should be considered when deciding which electricity system to use. However, other resources that provide more detailed analysis of these aspects should be considered as well. Our analysis here is by no means comprehensive and other factors related to socioeconomics, the environment, or safety that are not touched on here may also be important to consider.

## Limitations

Although we have put forth significant effort to ensure that our simulations correspond accurately to the situation in Estonia, it is important to recognize some of the potential limitations of this study.

One source of uncertainty is the behavior of the broader European electricity market, of which Estonia is a part. Estonia has electricity connections to Finland and Latvia and electricity is exchanged across the region. Modeling this complex network of interconnected local markets is beyond the scope of this study, and we made the simple assumption that up to 30% of the demand in Estonia could be exported at any given moment. This assumption is consistent with data for the year of 2018, when Estonia was still producing enough electricity to be a net exporter [58]. However, the export potential likely fluctuates over time. If more can be exported, especially when wind turbines are producing a lot of electricity, then Estonia could export more electricity and have fewer curtailments than in the simulation results shown here. The opposite would also occur if there is less demand for electricity abroad.

The offshore wind turbines may also have a higher capacity factor than we used for the simulations here. In these simulations we only had data for onshore turbines because no offshore turbines have so far been built in Estonia, and therefore, we assumed the same production profile for both onshore and offshore. Some sources indicate that the offshore wind turbines could actually have a higher capacity factor [4, 146], although it appears this is not always the case, which is why we did not assume a higher capacity factor for offshore in our simulations [5]. If a higher capacity factor could be achieved offshore, then a smaller wind turbine capacity would be needed and it would be more likely that a lower cost could be achieved.

Additionally, the estimated costs for CCS and small modular nuclear reactors are based on the assumption that other plants would have been built before with the same technology. However, many of these technologies are still new and if Estonia built one of the first plants using a given technology, then the cost would likely be higher than the estimates given here.

Many of these energy infrastructure projects also involve large upfront capital costs, and the costs and complexities of such large projects mean there is a risk of going significantly over time and over budget [161]. This has been the case with several recent large nuclear plants in Europe and America, such as the EPR reactors built at Olkiluoto in Finland and Flamanville in France [162, 163]. Similar overruns have also occurred with large CCS projects, such as the Kemper County IGCC plant in the United States [164–166]. Therefore, it is worth remembering that additional factors, such as poor project management, opposition from the public, or regulatory uncertainty, could drastically increase the cost for any of the proposed energy projects discussed here.

It is also worth noting that various strategies and technologies have been proposed for improving the flexibility of electricity systems with high levels of wind and solar, and these could in turn reduce the System LCOE [167]. Some ideas focus on using other infrastructure in the economy for additional energy storage, such as using the batteries of electric vehicles to help balance the grid [168]. Other strategies would improve interconnectivity. This could be done by building new transmission lines to allow better balancing of supply and demand between regions, or increasing the performance of existing lines [169, 170]. One such example for increasing the performance of existing lines is dynamic thermal rating. The capacity of transmission lines is limited to ensure that the temperature of the lines does not exceed their rated value. Currently, a fixed, static thermal rating is generally used, but by dynamically adjusting the thermal rating based on actual environmental conditions, the capacity of the lines could be increased during some periods [170]. This could reduce the need for and cost of grid improvements. Another often mentioned strategy is demand response, which involves enabling consumers to change their electricity usage based on the current state of their electricity market. For example, a consumer's air conditioner or water heater could be run when wind and solar are producing and the price is cheaper [171]. Many of these strategies and technologies are still being developed and the economic feasibility of them is generally still unsure, and therefore, we did not include any of them in our simulations.

More generally, all estimates have uncertainty associated with them. We have quantified this uncertainty to the best of our ability by using a Monte Carlo simulation method. Using this method, parameter values are selected from the likely range of values, and repeating the simulation with a range of parameters gives us an estimate of how much the results might vary due to the uncertainty in the underlying model parameters. That said, the situation could change. For instance, the demand in Estonia might not follow historical trends. Also, the cost of various technologies will likely change over time. Furthermore, in this study, we relied often on literature estimates for cost and other parameters. Better accuracy could be achieved by performing more detailed engineering designs specific to Estonia and making estimates using those.

On the most basic level, there are simply a variety of perspectives and approaches that can be used to evaluate an electricity system. Here we evaluated the grid at a system-wide level and focused on how demand, production, storage, and the balance between them vary in time. There are, however, other perspectives that could and should be considered as well. For instance, one could also look at these different scenarios in terms of grid operation and control because the behavior of some configurations may be more difficult to predict and manage. In addition, safety and environmental impacts should be considered. For instance, it has been shown that fossil fuels have much higher external costs (i.e. health and environmental impacts)

than other energy production methods [62, 155–157, 159, 172, 173]. In this study, we also did not look at social aspects, such as whether or not workers losing their jobs if oil shale plants closed would be able to take advantage of the jobs created by newer energy technologies. Although the results in this study are important to consider before choosing which strategy to pursue to reduce emissions, other perspectives and approaches should also be taken into account.

## Conclusions

Our simulations showed that to achieve the lowest possible LCOE Estonia should complement wind turbines with some sort of dispatchable power plant to provide stability to its electricity system. Using energy storage to provide stability is not a viable option in the foreseeable future because it is significantly more expensive. Our calculations for underground pumped hydro storage, which is one of the cheapest storage technologies, showed that the LCOE would be about 40–50% higher than a comparable system with a power plant. Based on literature estimates, other storage technologies, such as hydrogen storage or batteries, would cost even more. Energy storage might be useful on a smaller scale to provide important ancillary services to the grid, but at the scale required to stabilize an entire grid of wind turbines and solar panels, it is currently too expensive.

Stable dispatchable power could be provided by using oil shale, pyrolysis gas, or nuclear power plants, but using oil shale with CCS is more expensive than the other options. A small modular nuclear power plant (Generation III+) is expected to have a lower LCOE than an oil shale plant with CCS. Utilizing pyrolysis gas would likely have the lowest cost, but pyrolysis gas is a byproduct and is not available in sufficient quantities to provide the approximately 600 $MW_e$ of dispatchable power needed to stabilize the grid. Therefore, an oil shale or nuclear power plant would still be needed. Simulations also indicated that using nuclear power could reduce the amount of electricity overproduction. This is mainly due to the fact that nuclear power plants can be operated more flexibly than oil shale power plants, unless specific modifications are made to the oil shale boilers to improve their flexibility.

Additionally, although CCS significantly reduces $CO_2$ emissions, an oil shale power plant with CCS would still have probably an order of magnitude higher greenhouse gas emissions over its lifetime than a nuclear plant. Statistically, oil shale power plants also cause significantly more deaths than nuclear power plants, mainly due to air pollution.

Note also that all of the potential low-emission scenarios are expected to have a higher LCOE than the current wholesale price of electricity in the Nordpool market. This indicates that the market may need to be adjusted to compensate power plants or energy storage for the additional services they provide to the grid, such as enabling flexible production and ensuring a reserve capacity. Payment for such services could give these energy projects an additional stable revenue stream, which would be important for securing the large investment capital needed for power plants or large-scale energy storage projects.

In summary, transitioning to low-emission electricity production will most likely increase the cost of electricity in Estonia. However by balancing wind turbines with stable production from a smaller number of thermal power plants the country can achieve the transition at a lower cost.

## Author Contributions

**Conceptualization:** Zachariah Steven Baird, Dmitri Neshumayev, Oliver Järvik.

**Formal analysis:** Zachariah Steven Baird.

**Investigation:** Zachariah Steven Baird.

**Methodology:** Zachariah Steven Baird, Dmitri Neshumayev, Oliver Järvik, Kody M. Powell.

**Software:** Zachariah Steven Baird.

**Validation:** Dmitri Neshumayev, Oliver Järvik, Kody M. Powell.

**Visualization:** Zachariah Steven Baird.

**Writing – original draft:** Zachariah Steven Baird.

**Writing – review & editing:** Zachariah Steven Baird, Dmitri Neshumayev, Oliver Järvik, Kody M. Powell.

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
