## [Decision Letter · Decision Letter 0]

2 Nov 2021

PONE-D-21-32809Comparison of the most likely low-emission electricity production systems in EstoniaPLOS ONE

Dear Dr. Baird,

Thank you for submitting your manuscript to PLOS ONE. After careful consideration, we feel that it has merit but does not fully meet PLOS ONE’s publication criteria as it currently stands. Therefore, we invite you to submit a revised version of the manuscript that addresses the points raised during the review process.

We look forward to receiving your revised manuscript.

Kind regards,

Jiashen Teh

Academic Editor

PLOS ONE

Journal Requirements:

"The Estonian Research Council funded this research under the National Programme for

Addressing Socio-Economic Challenges through R&D (RITA), which is supported by the

Estonian Government and European Regional Development Fund, under the project

“Climate Change Mitigation with CCS and CCU Technologies” (ClimMit, grant No.

RITA1/02-20)."

"The Estonian Research Council (https://www.etag.ee/en/) funded this research under the National Programme for Addressing Socio-Economic Challenges through R&D (RITA), which is supported by the Estonian Government and European Regional Development Fund, under the project “Climate Change Mitigation with CCS and CCU Technologies” (ClimMit, grant No. RITA1/02-20). The funders had no role in study design, data collection and analysis, decision to publish, or preparation of the manuscript."

4. Please ensure that you refer to Figure 2 in your text as, if accepted, production will need this reference to link the reader to the figure.

Additional Editor Comments:

The reviewers are mainly concerned with the references used and pointed out the lack comparison studies in the manuscript. The authors are advised to look into them very carefully.

Reviewers' comments:

Reviewer's Responses to Questions

**Comments to the Author**

1. Is the manuscript technically sound, and do the data support the conclusions?

Reviewer #1: Yes

Reviewer #2: Yes

2. Has the statistical analysis been performed appropriately and rigorously? 

Reviewer #1: Yes

Reviewer #2: Yes

3. Have the authors made all data underlying the findings in their manuscript fully available?

Reviewer #1: Yes

Reviewer #2: Yes

4. Is the manuscript presented in an intelligible fashion and written in standard English?

Reviewer #1: Yes

Reviewer #2: Yes

5. Review Comments to the Author

Reviewer #1: The authors presented a study on comparing the electrical systems of Estonia. However, I think the following can help to improve the quality of the paper further

1) Please include some discussion on the use of the DTR system for improving the transmission capacity of the power systems and provide a descriptive comparison on how this DTR system is beneficial? Some example studies for reference are:

["Reliability impact of dynamic thermal rating system in wind power integrated network", IEEE Trans Reliability],

["Prospects of using the dynamic thermal rating system for reliable electrical networks: A review", IEEE Access],

["Reliability impacts of the dynamic thermal rating and battery energy storage systems on wind-integrated power networks", IEEE Access],

["Reliability impacts of the dynamic thermal rating system on smart grids considering wireless communications". IEEE Access],

["Risk-based management of transmission lines enhanced with the dynamic thermal rating system", IEEE Access],

["Probabilistic Peak Demand Matching by Battery Energy Storage Alongside Dynamic Thermal Ratings and Demand Response for Enhanced Network Reliability", IEEE Access], ["Network topology optimisation based on dynamic thermal rating and battery storage systems for improved wind penetration and reliability", Applied energy],

["Demand Response and Dynamic Line Ratings for Optimum Power Network Reliability and Ageing", IEEE Access],

["Fuzzy Dynamic Thermal Rating System based SIPS for Enhancing Transmission Line Security", IEEE Access]

Reviewer #2: All factors and parameters with the respective impact should be highlighted again after all analysis have been done. A table of graph can be included to strengthen your findings. I am not satisfy with too many references. The authors should be focused on recent journals (5 years back).

6. PLOS authors have the option to publish the peer review history of their article (what does this mean?). If published, this will include your full peer review and any attached files.

Reviewer #1: No

Reviewer #2: No

---

## [Author Response · Author response to Decision Letter 0]

29 Nov 2021

Response to reviewers

Reviewer 1

Please include some discussion on the use of the DTR system for improving the transmission capacity of the power systems and provide a descriptive comparison on how this DTR system is beneficial?

 Although dynamic thermal ratings is an interesting and important topic, it is only one of the many potential technologies and techniques that will be needed in a grid with large amounts of wind and solar. The grid improvement costs, which are the cost component that could be reduced with DTR, are only a minor factor in the overall system cost that was the focus of this article, accounting for only about 1-2 EUR/MWh. The impact of DTR on the overall system cost would be even smaller because some grid improvements cannot be avoided using DTR. The current article takes a higher-level approach and focuses on the Estonian electricity system as a whole, and therefore, including a discussion of all potential technologies and implementation details is beyond the scope of our current article. We do, however, thank the reviewer for the informative references and may find them useful in our future work.

Reviewer 2

All factors and parameters with the respective impact should be highlighted again after all analysis have been done. A table of graph can be included to strengthen your findings.

 We have now included a sensitivity analysis that investigates the impact of all the variable parameters used in the model. Figure 8 gives an overview of this analysis and a new subsection was added to discuss the results.

I am not satisfy with too many references. The authors should be focused on recent journals (5 years back).

 It is true that the article has many references, but each of these references are important to the article and no unnecessary references have been included. One reason there are so many references is because this study required estimates for many different parameters used in modeling the electricity system in Estonia.

 We have also used the most recent studies we could find that give the information we needed. A large portion of the references are actually to literature published within the past 5 years. Of the remaining references, the majority have at least been published within the last 10 years. For some data and information, it was not possible to find more recent sources. Additionally, the underlying goal when selecting literature is to find the most reliable information. Although more recent literature may be, on average, more likely to be reliable, this is not always the case and sometimes citing an older, but more reliable, study is best. This is one reason we have cited some sources that are older than 10 years: they are key studies in the field and provide reliable data. One such example is the IPCC report on carbon capture published in 2005.

 For these reasons, we have kept the list of cited works the same, but if the reviewer sees specific references that could be removed or replaced with more recent studies, we would be willing to take those suggestions.

Editor

 We have renamed the figure files and reformatted the article to meet PLOS ONE's style guidelines to the best of our knowledge.

We note that the grant information you provided in the ‘Funding Information’ and ‘Financial Disclosure’ sections do not match.

 We have updated this information so it matches.

Funding information should not appear in the Acknowledgments section or other areas of your manuscript.

 This was removed.

Please ensure that you refer to Figure 2 in your text as, if accepted, production will need this reference to link the reader to the figure.

 Thank you for noticing this error. We had simply made a mistake and referenced Figure 1 in the spot where Figure 2 should have been referenced. We have corrected this.

Please review your reference list to ensure that it is complete and correct.

 We have reviewed our list of references and to the best of our knowledge all are correct.

---

## [Decision Letter · Decision Letter 1]

7 Dec 2021

PONE-D-21-32809R1Comparison of the most likely low-emission electricity production systems in EstoniaPLOS ONE

Dear Dr. Zachariah Baird

Thank you for submitting your manuscript to PLOS ONE. After careful consideration, we feel that it has merit but does not fully meet PLOS ONE’s publication criteria as it currently stands. Therefore, we invite you to submit a revised version of the manuscript that addresses the points raised during the review process.

We look forward to receiving your revised manuscript.

Kind regards,

Jiashen Teh

Academic Editor

PLOS ONE

Journal Requirements:

Reviewers' comments:

Reviewer's Responses to Questions

**Comments to the Author**

1. If the authors have adequately addressed your comments raised in a previous round of review and you feel that this manuscript is now acceptable for publication, you may indicate that here to bypass the “Comments to the Author” section, enter your conflict of interest statement in the “Confidential to Editor” section, and submit your "Accept" recommendation.

Reviewer #1: (No Response)

Reviewer #2: All comments have been addressed

2. Is the manuscript technically sound, and do the data support the conclusions?

Reviewer #1: Yes

Reviewer #2: Yes

3. Has the statistical analysis been performed appropriately and rigorously? 

Reviewer #1: Yes

Reviewer #2: Yes

4. Have the authors made all data underlying the findings in their manuscript fully available?

Reviewer #1: Yes

Reviewer #2: Yes

5. Is the manuscript presented in an intelligible fashion and written in standard English?

Reviewer #1: Yes

Reviewer #2: Yes

6. Review Comments to the Author

Reviewer #1: The authors presented a study on comparing the electrical systems of Estonia. However, I think the following can help to improve the quality of the paper further 1) Please include some discussion on the use of the DTR system for improving the transmission capacity of the power systems and provide a descriptive comparison on how this DTR system is beneficial? Some example studies for reference are: ["Reliability impact of dynamic thermal rating system in wind power integrated network", IEEE Trans Reliability], ["Prospects of using the dynamic thermal rating system for reliable electrical networks: A review", IEEE Access], ["Reliability impacts of the dynamic thermal rating and battery energy storage systems on wind-integrated power networks", IEEE Access], ["Reliability impacts of the dynamic thermal rating system on smart grids considering wireless communications". IEEE Access], ["Risk-based management of transmission lines enhanced with the dynamic thermal rating system", IEEE Access], ["Probabilistic Peak Demand Matching by Battery Energy Storage Alongside Dynamic Thermal Ratings and Demand Response for Enhanced Network Reliability", IEEE Access], ["Network topology optimisation based on dynamic thermal rating and battery storage systems for improved wind penetration and reliability", Applied energy], ["Demand Response and Dynamic Line Ratings for Optimum Power Network Reliability and Ageing", IEEE Access], ["Fuzzy Dynamic Thermal Rating System based SIPS for Enhancing Transmission Line Security", IEEE Access]

Reviewer #2: The authors have addressed all comments in previous session. The presented manuscript remained the list of references. But with the given justifications, i would accept them. The manuscript is good for publication.

7. PLOS authors have the option to publish the peer review history of their article (what does this mean?). If published, this will include your full peer review and any attached files.

Reviewer #1: No

Reviewer #2: No

---

## [Author Response · Author response to Decision Letter 1]

9 Dec 2021

Response to reviewers

Reviewer 1

Comment: The authors presented a study on comparing the electrical systems of Estonia. However, I think the following can help to improve the quality of the paper further 1) Please include some discussion on the use of the DTR system for improving the transmission capacity of the power systems and provide a descriptive comparison on how this DTR system is beneficial?

Response: We have included the following discussion of DTR systems in the revised version of the article:

"One such example for increasing the performance of existing lines is dynamic thermal rating. The capacity of transmission lines is limited to ensure that the temperature of the lines does not exceed their rated value. Currently, a fixed, static thermal rating is generally used, but by dynamically adjusting the thermal rating based on actual environmental conditions, the capacity of the lines could be increased during some periods."

To round out the discussion and improve the flow of the text, we included these sentences in a larger paragraph that discusses this and a few other proposed strategies that could be used to improve the flexibility and reduce the cost of grids with a high penetration of renewables.

 Earlier Reviewer 2 expressed concern about the number of references in the article. Obviously, introducing this new section increased the number of references. To also accommodate the earlier comments of Reviewer 2, we have limited the number of new references we included and gave preference to sources published within the last 5 years. And as requested in the requirements of the journal, here is a list of the new references we have added (references 168-172 in the references list):

168. Cochran J, Miller M, Zinaman O, Milligan M, Arent D, Palmintier B, O’Malley M, Mueller S, Lannoye E, Tuohy A, Kujala B, Sommer M, Holttinen H, Kiviluoma J, Soonee SK. Flexibility in 21st Century Power Systems [Internet]. National Renewable Energy Lab. (NREL), Golden, CO (United States); 2014 May [cited 2021 Dec 8]. Report No.: NREL/TP-6A20-61721. Available from: https://www.osti.gov/biblio/1130630

169. Jian L, Zechun H, Banister D, Yongqiang Z, Zhongying W. The future of energy storage shaped by electric vehicles: A perspective from China. Energy. 2018;154:249–57. 

170. Martinot E. Grid Integration of Renewable Energy: Flexibility, Innovation, and Experience. Annu Rev Environ Resour. 2016;41(1):223–51. 

171. Karimi S, Musilek P, Knight AM. Dynamic thermal rating of transmission lines: A review. Renew Sustain Energy Rev. 2018 Aug 1;91:600–12. 

172. Verma R. Application of Computational Techniques in Demand Response: A Review. In: 2021 7th International Conference on Advanced Computing and Communication Systems (ICACCS). 2021. p. 1242–6.

---

## [Editor Report · Decision Letter 2]

10 Dec 2021

Comparison of the most likely low-emission electricity production systems in Estonia

PONE-D-21-32809R2

Dear Dr. Baird,

We’re pleased to inform you that your manuscript has been judged scientifically suitable for publication and will be formally accepted for publication once it meets all outstanding technical requirements.

Kind regards,

Jiashen Teh

Academic Editor

PLOS ONE
---

## [Editor Report · Acceptance letter]

17 Dec 2021

PONE-D-21-32809R2 

Comparison of the most likely low-emission electricity production systems in Estonia 

Dear Dr. Baird:

I'm pleased to inform you that your manuscript has been deemed suitable for publication in PLOS ONE. Congratulations! Your manuscript is now with our production department. 

Kind regards, 

on behalf of

Dr. Jiashen Teh 

Academic Editor

PLOS ONE